# Record sea surface temperature jump in 2023–2024 unlikely but not unexpected

Jens Terhaar[1,2✉], Friedrich A. Burger[1,2], Linus Vogt[3], Thomas L. Frölicher[1,2] & Thomas F. Stocker[1,2]

Global ocean surface temperatures were at record levels for more than a year from April 2023 onwards, exceeding the previous record in 2015–2016 by 0.25 °C on average between April 2023 and March 2024[1]. The nearly global extent and unprecedented intensity of this event prompted questions about how exceptional it was and whether climate models can represent such record-shattering jumps in surface ocean temperatures[2]. Here we construct observation-based synthetic time series to show that a jump in global sea surface temperatures that breaks the previous record by at least 0.25 °C is a 1-in-512-year event under the current long-term warming trend (1-in-205-year to 1-in-1,185-year event; 95% confidence interval). Without a global warming trend, such an event would have been practically impossible. Using 270 simulations from a wide range of fully coupled climate models, we show that these models successfully simulate such record-shattering jumps in global ocean surface temperatures, underpinning the models' usefulness in understanding the characteristics, drivers and consequences of such events. These model simulations suggest that the record-shattering jump in surface ocean temperatures in 2023–2024 was an extreme event after which surface ocean temperatures are expected to revert to the expected long-term warming trend.

Since April 2023, global ocean (60° S–60° N; excluding cloudy and largely sea-ice covered polar regions owing to sparse data) surface temperatures have exceeded previous sea surface temperature (SST) records by a large margin (Fig. 1a). This record-breaking SST event is not only a global record-breaking event but also unprecedented in the magnitude by which it surpassed previous records. Over summer 2023, the margin by which SSTs exceeded the previous record that occurred in 2015–2016 increased to 0.2–0.3 °C (Fig. 1a). Overall, the annually averaged SSTs from April 2023 to March 2024 based on NOAA Optimum Interpolation (OI) Sea Surface Temperature (SST) V2.1 (NOAA OISST V2.1)[1] were 0.25 °C larger than the previous record SSTs when averaged over the same months of the year. This global record-shattering jump in SSTs ('record-shattering jump' is here defined as a record-breaking jump in annual (April to March) and globally averaged SSTs that exceeds previous records by at least 0.25 °C, as observed in 2023–2024) coincides in time with record atmospheric surface temperatures in late 2023[3–6] and early 2024[7,8]. Moreover, the record-shattering jump in SSTs is believed to be responsible for the global atmospheric record surface warming[5,6], although surface temperature extremes over land and ocean are not necessarily related[9,10]. The record-shattering jump in globally averaged SSTs has been a subject of much attention in the scientific community[11,12] and the general public[13,14]. It has, for example, recently been argued that part of the jump was caused by low albedo owing to reduced low-cloud cover[15]. Since mid-July 2024, globally averaged SSTs are no longer record-breaking but still remain warmer than in any year before the jump in 2023 (Fig. 1a).

Large increases in SST, which locally manifest as marine heatwaves, can affect regional climate patterns[16] and sea–air carbon dioxide fluxes[17], and substantially impact the marine environment[18]. For example, marine heatwaves in the Indian Ocean can influence monsoon wind and precipitation over India, affecting water and food security[19]. They can also interact with and intensify tropical cyclones[20], increasing their destructiveness. The biological impacts of marine heatwaves include mass die-offs of invertebrates, fish, birds and marine mammals[21,22], coral bleaching[23], declines in key species, and complete ecosystem restructuring[24,25], all of which have socioeconomic consequences[26,27].

Although record-breaking jumps in global SSTs occur when a long-term warming trend is superimposed onto an exceptionally warm year owing to climate variability[28], the record-shattering jump in globally averaged SSTs from April 2023 to March 2024 has broken previous records by a substantially larger margin than previous record-breaking jumps in SSTs. The three largest margins by which globally averaged SSTs previously broke records were 0.16 °C in 2015–216, 0.14 °C in 1997–1998 and 0.09 °C in 2009–2010. Owing to the unprecedented margin in the record-breaking jump in SSTs in 2023–2024, this jump in SSTs came as a surprise for the public and the scientific community. This event has raised questions about the likelihood of such a jump and whether jumps of this size are simulated in climate models[2]. The failure of state-of-the-art climate models to reproduce events such as the jump in SSTs in 2023–2024 would consequently question the ability of these models to assess future risks associated with anthropogenic climate change[29]. However, if climate models are able to simulate such record-shattering jumps in globally averaged SSTs, they would

[1]Climate and Environmental Physics, Physics Institute, University of Bern, Bern, Switzerland. [2]Oeschger Center for Climate Change Research, University of Bern, Bern, Switzerland. [3]LOCEAN/IPSL, Sorbonne Université, CNRS, IRD, MNHN, Paris, France. ✉e-mail: jens.terhaar@unibe.ch

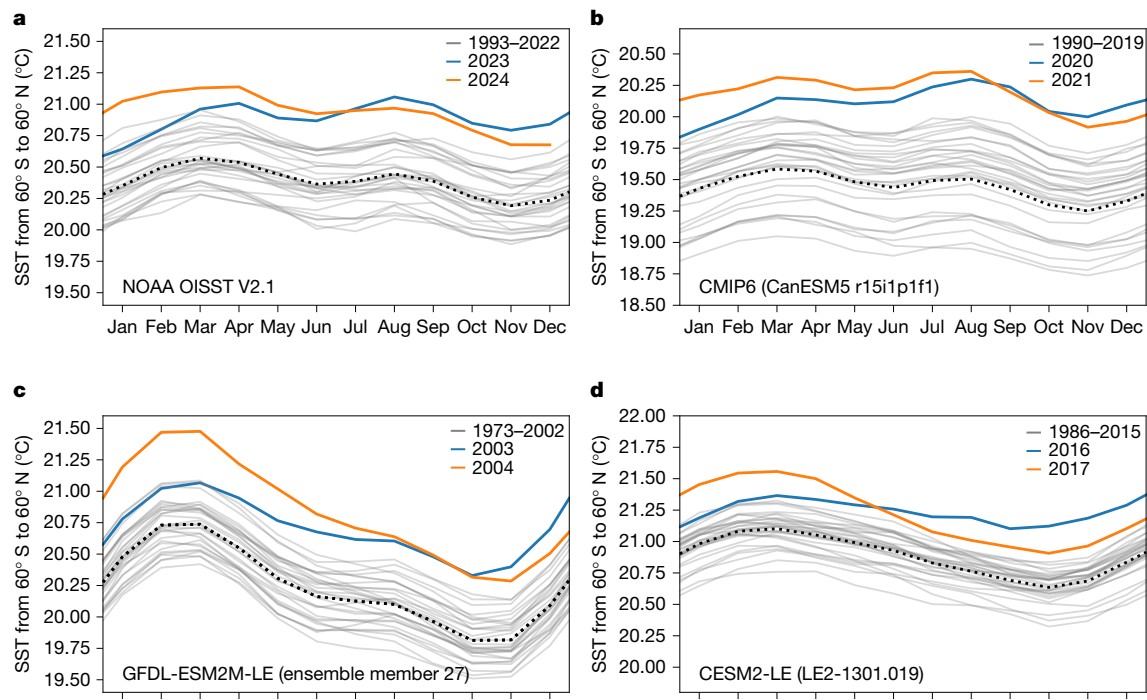

**Fig. 1 | Record-shattering jumps in global SSTs as observed in 2023–2024 also occur in climate model simulations. a–d**, Monthly mean SST anomalies for the largest record-shattering annual (April to March) global (60° S–60° N) SST events before 2024 for observations from NOAA OISST V2.1[1] (**a**), and climate model simulations from one CMIP6 simulation (CanESM5[39,40] r15i1p1f1; **b**), from one simulation of the GFDL-ESM2M[41] large ensemble (LE)[42] (ensemble member 27; **c**) and from one simulation of the CESM2[43] large ensemble[44] (ensemble member LE2-1301.019; **d**). The years of the onset of the respective events are shown as blue lines, the years of the subsequent decline are shown as orange lines, and the 30 preceding years are shown as grey lines with their mean as a black dotted line. For each of the three climate model groups (Coupled Model Intercomparison Project Phase 6 (CMIP6), GFDL-ESM2M-LE and CESM2-LE), the largest record-breaking global jump in SSTs before 2023 is shown. Monthly SST anomalies for all simulated record-shattering global jumps in SSTs between 2000 and 2040 that are larger in magnitude than the observed global jump in SSTs in 2023 and 2024 are shown in Extended Data Fig. 1.

deliver analogues to study the evolution of the ongoing event and to see whether temperatures decrease again or remain high. In addition, the climate models could be used to identify the drivers of such record-shattering jumps in SSTs, and their consequences for other parts of the climate system and the marine ecosystem.

## Return period of record-shattering SST jumps

Here we use observation-based monthly mean SST estimates from various data and reanalysis products (NOAA OISST V2.1[1], ERA5[30], The Hadley Centre Global Sea Ice and Sea Surface Temperature (HadISST)[31] and Extended Reconstructed Sea Surface Temperature (ERSST)[32]) to quantify the likelihood of the global SST jump in 2023–2024 and assess whether it could have occurred without anthropogenic warming. Owing to the relatively short observational SST record, return periods of rare extreme events, such as the record-shattering jump in globally averaged SSTs observed in 2023–2024, cannot be directly inferred from that observational record. To quantify the likelihood of a record-shattering jump in globally averaged SSTs that exceeded the last record by at least 0.25 °C, as in 2023–2024, we constructed synthetic time series of 100 million years using an autoregressive model of order one (AR(1)) using observation-based estimates of the trend, autocorrelation and standard deviation of annual globally averaged SSTs with respective uncertainties (see Methods for a detailed description of how these values and their uncertainties are quantified).

Based on these synthetic observation-based time series, the record-shattering global jump in 2023–2024 was a 1-in-512-year event (mean estimate) under the current long-term warming trend (1-in-205-year to 1-in-1,185-year event; 95% confidence interval based on uncertainties of the observation-based trend, standard deviation and autocorrelation estimates; Fig. 2 and Methods). This result is qualitatively insensitive to the choice of the autoregressive model, to the methods that are used to estimate the trend, the autocorrelation and the standard deviation of annual globally averaged SSTs, as well as to the observation-based SST dataset that is used for the analysis (Methods). Without underlying warming, a record-shattering jump as observed in 2023–2024 is practically impossible. We found indeed no record-shattering jumps in our synthetic time series without a long-term warming trend, irrespective of the variability or autocorrelation characteristics (Extended Data Fig. 2).

## Record-shattering SST jumps in climate models

Having quantified an observation-based estimate of the return period of such record-shattering jumps in globally averaged SSTs, we show that such jumps—exceeding the previous record by at least 0.25 °C— were simulated 11 times across 270 simulations from 35 different state-of-the-art climate models (Methods) between 2000 and 2040. These four decades encompass the time when the record-shattering SST jump occurred in the real world (Fig. 3a and Extended Data Fig. 1). As the 270 climate model simulations here comprise a total of 11,070 years, the likelihood for a single year to experience a record-shattering global jump in SSTs as observed in 2023–2024 in the climate models is 0.1%, making the record-shattering SST event in 2023–2024 a 1-in-1,006-year event in climate models (1-in-563-year to 1-in-2,016-year event; 95% confidence interval using the Pearson–Clopper confidence interval for binomial experiments; Methods and Fig. 3b). Although the return period estimate based on climate

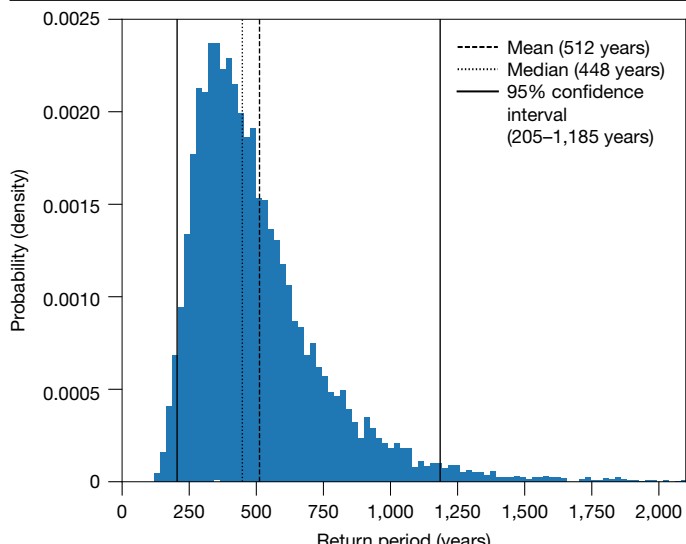

**Fig. 2 | The return period of record-shattering jumps in annually averaged global SSTs based on past observed SSTs.** Observation-based estimate of the probability density of the return period of record-shattering jumps in synthetic time series of global SSTs that break the previous record by at least 0.25 °C. The mean return period (dashed black line), the median return period (dotted black line) and the limits of the 95% confidence interval (solid black lines) are also shown (see Methods for details).

models is approximately twice as large as the observation-based estimate of the return period, it lies within the confidence interval of the observation-based estimate and the confidence intervals of both estimates largely overlap. The difference in the return period might be owing to lower warming trends and higher autocorrelations in the models compared with the observation-based estimates (Methods). However, the observation-based estimate of the return period is also highly uncertain as it is difficult to estimate the real trend, variability and autocorrelation over a short time series that is strongly influenced by natural climate variability so that the return period of such events might well be 1,006 years as estimated from the models. Overall, the probability of a simulated record-breaking jump in SSTs with a certain magnitude decreases if that magnitude increases. For example, jumps that exceed 0.2 °C are 6–7 times more likely than jumps exceeding

0.25 °C, whereas jumps that exceed the previous record by a larger magnitude than 0.25 °C become less likely but are not impossible in climate model simulations (Fig. 3b). Under high-emissions scenarios, the probability of record-shattering events increases (Extended Data Fig. 3a), in line with higher rates of warming. However, under strong mitigation scenarios, the warming trend reduces in the future and there are no simulated record-shattering events simulated across the model ensemble (Extended Data Fig. 3b).

Record-shattering global jumps in SSTs (>0.25 °C) are simulated by different models with a wide range of trends, autocorrelations and variabilities. It is thus not only the so-called hot models[33] with high transient climate responses, such as IPSL-CM6A-LR, CESM2 and CanESM5, that simulate such events. Instead, models with a small transient climate response and warming trend, such as GFDL-ESM2M and MIROC6, also produce these record-shattering jumps as variability and autocorrelation also affect the return period, and not just the long-term trend (Extended Data Fig. 2). Overall, no model stands out with an unusually small or high number of extreme events per number of simulated years (Methods).

## Pattern of record-shattering SST jumps

The record-shattering global SST anomalies observed in 2023–2024 (data from NOAA OISST V2.1[1]) were especially pronounced in the North Atlantic, Eastern Tropical Pacific and North Pacific (Fig. 4a). Although all three regions (see exact definitions in Methods) show high regional SST anomalies compared with the previous 30 years, only the North Atlantic SST anomalies have broken previous records. The North Atlantic SSTs in 2023–2024 surpassed the previous record by 0.42 °C, which was 0.17 °C more than the margin by which the global SSTs had surpassed the previous record in the same year. In the North Pacific, SST anomalies were only once larger than in 2023–2024 and in the Eastern Tropical Pacific, SST anomalies were only twice larger than in 2023–2024. In climate models, all 11 jumps in globally averaged SSTs that are as large as that in 2023–2024 coincide with an El Niño event, that is, a positive El Niño–Southern Oscillation phase of at least 1.5 °C in the El Niño 3.4 index (Fig. 4b–d). As the three most recently observed record-breaking jumps that broke the respective previous record in globally averaged SSTs by unprecedented margins (2023–2024, 2015–2016 and 1997–1998) also occurred during a positive El Niño phase (1.3 °C, 1.9 °C and 1.7 °C respectively), a strong El Niño appears to be a necessary, but not sufficient, condition for such an event.

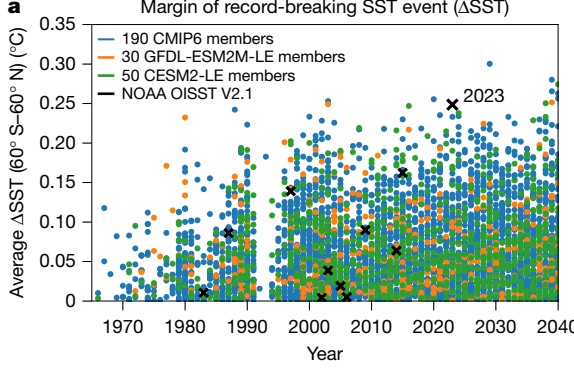

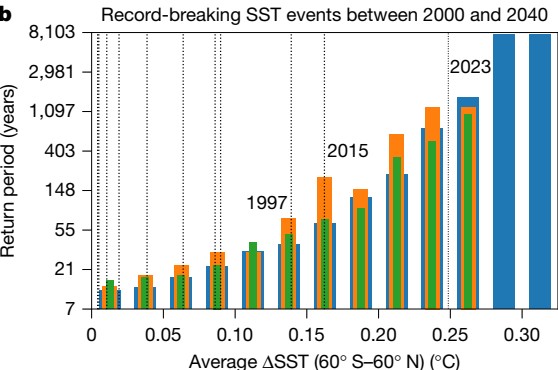

**Fig. 3 | The return period of record-breaking annual SST events increases with the size of the margin by which they exceed previous records. a**, Record-breaking annual (April–March) global (60° S–60° N) SST events and the margin (ΔSST) by which they exceeded previous records based on 190 climate model simulations from CMIP6 (blue dots) (Extended Data Table 1), 30 members of the GFDL-ESM2M-LE[41,42] (orange dots), 50 members from the CESM2-LE[43,44] (green dots), and SST observation-based estimates from NOAA OISST V2.1[1] (black crosses). **b**, Return periods of record-breaking events of a given magnitude (binned in regular bins of 0.025 °C) calculated based on their occurrence in **a**

for the ensembles of CMIP6 (blue), GFDL-ESM2M (orange) and CESM2 (green). The return periods indicate the return period for a record-breaking event of that magnitude and not of an event that is equal or larger than the magnitude of that bin. Return periods for ΔSSTs of more than 0.275 °C are not simulated by the large ensembles of GFDL-ESM2M and CESM2, possibly owing to their relatively small sample size (1,230 years between 2000 and 2040 for the GFDL-ESM2M-LE and 2,050 years for the CESM2-LE). Observed record-breaking events, corresponding to the crosses in **a**, are shown as dotted black vertical lines.

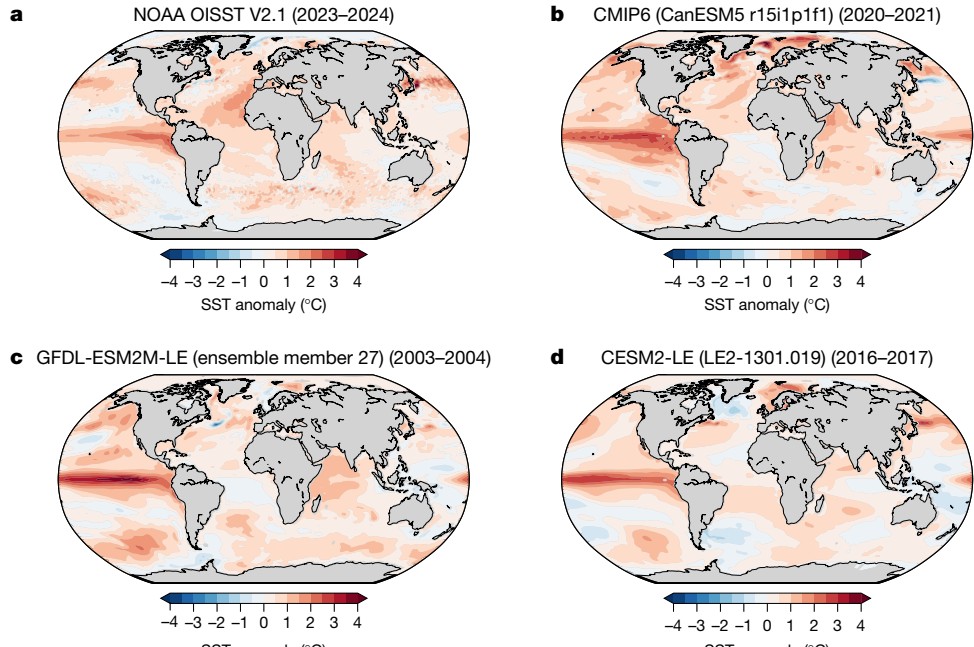

**a** NOAA OISST V2.1 (2023–2024)

SST anomaly (°C)
−4 −3 −2 −1 0 1 2 3 4

**b** CMIP6 (CanESM5 r15i1p1f1) (2020–2021)

SST anomaly (°C)
−4 −3 −2 −1 0 1 2 3 4

**c** GFDL-ESM2M-LE (ensemble member 27) (2003–2004)

SST anomaly (°C)
−4 −3 −2 −1 0 1 2 3 4

**d** CESM2-LE (LE2-1301.019) (2016–2017)

SST anomaly (°C)
−4 −3 −2 −1 0 1 2 3 4

**Fig. 4 | Anomalies of record-shattering jumps in SSTs in observations and climate models are mostly localized in the equatorial Pacific, North Pacific and North Atlantic. a**–**d**, SST anomalies for record-shattering (record-breaking events of largest magnitude before 2024) annual (April–March) global (60° S–60° N) SST events for observations from NOAA OISST V2.1[1] (**a**), and the three climate model groups in Fig. 1: CMIP6 (CanESM5[39,40] r15i1p1f1; **b**), the GFDL-ESM2M-LE[41,42] (ensemble member 27; **c**) and the CESM2-LE[43,44] (ensemble member LE2-1301.019; **d**). The anomalies are calculated based on the average SSTs of the 30 years preceding the respective record-shattering event at the years corresponding to those presented in Fig. 1 for the member of the respective climate model group.

For the 11 simulated global record-shattering jumps in SSTs, record-breaking regional SST anomalies occur 10 times in the Eastern Tropical Pacific, 7 times in the North Atlantic and 5 times in the North Pacific. The largest margin by which North Atlantic SSTs records are broken during the 11 global record-shattering jumps in SSTs in climate models is 0.33 °C, a margin that is 0.09 °C smaller than the extraordinarily large observed margin in 2023–2024. This observed jump in the North Atlantic was probably caused by a high level of surface solar radiation, weak winds, tropical air[34] and changes in low-cloud cover[15]. Such large jumps in North Atlantic SSTs, as observed in 2023–2024, are identified five times in the 11,070 model years simulated between 2000 and 2040. However, coincident global and regional record-shattering jumps at the observed magnitude of 2023–2024 are not found in the 11,070 model years between 2000 and 2040. An even larger model ensemble would be necessary to see whether climate models can simulate the combination of a global record-shattering jump in SSTs and a record-shattering jump in SSTs in the North Atlantic in 2023 and 2024. A sudden drop in aerosols from shipping emissions, which has not been prescribed as input to the climate models in the CMIP scenarios, might have contributed to the extremely large observed North Atlantic temperature anomaly[35].

## SST evolution after record-shattering jumps

We now use these model simulations to understand how SST anomalies typically develop over the years that follow a record-shattering jump in SSTs. In all simulations, the global SST anomalies stop being record-breaking, that is, they fall below previously measured temperatures in the same month, between May and October of the second year of the record SST anomalies, that is, 13 to 18 months after the beginning of the jump in globally averaged SSTs. In the real world, global SSTs dropped below record levels in July 2024 (Fig. 1a), 15 months after the beginning of the record-shattering jump in globally averaged SSTs. Furthermore, SST anomalies in climate models return to their level before the jump (grey lines in Fig. 1) between September in the year after the jump in SSTs started and September in the following year in 8 of the 11 simulated jumps (Extended Data Fig. 4). However, in 3 of the 11 events, the SST anomalies do not return to pre-jump levels over the next 10 years (Extended Data Fig. 4b–d) and beyond (Extended Data Fig. 5), so that the global SSTs have indeed permanently risen to a higher level in these simulations. However, even in these 3 cases, SSTs revert back to the expected long-term warming trend over the course of at most 8 years and do not shift to a new higher or steeper warming trajectory (Extended Data Fig. 5). The three cases occur in CanESM5 and IPSL-CM6A-LR, two climate models with extremely high transient climate responses[36] outside the recently assessed range[37] and with atmospheric warming rates from 1981 to 2014 that substantially exceed the observed rate[36]. As atmospheric warming and sea surface warming are generally strongly linked[38], the overly high SST warming rates facilitate continuing high levels of SST anomalies after the record-shattering jump in SSTs. Given that a return to pre-jump temperatures fails to occur in only these 'hot models'[33], it is likely that the global SST anomalies in the real world will return to temperatures more typical of those before the record-shattering jump by September 2025. If, however, observed SSTs do not return to pre-jump levels by September 2025, we expect SSTs to revert back towards the long-term trend within a few years (Extended Data Fig. 5). If this were not the case, the ongoing extreme event would not be consistent with climate model simulations.

## Outlook on warming rate and climate models

Based on long synthetic time series with temporal characteristics that match the available observations, we estimated that the observed record-shattering jump in globally averaged SSTs in 2023–2024 was a 1-in-a-512-year extreme event (1-in-205-year to 1-in-1,185-year event; 95% confidence interval) based on current warming rates. Such a jump would not have been possible without anthropogenic warming. We have further shown that climate models indeed simulate such global

(60° S–60° N) annual record-shattering jumps in SSTs that exceed the previous records by at least 0.25 °C, like the global jump in SSTs that was observed in 2023–2024. Moreover, the estimated return period of these events in climate models (1-in-a-1,006-year events in models) is within the confidence interval of the observation-based estimate of the return period. Furthermore, in these models, the simulated SST anomalies drop below record levels between May and September in the year after the jump in SSTs had started, consistent with the time when observed globally averaged SSTs stopped being record-breaking (July 2024). On the basis of the simulated record-shattering jumps, we conclude that it is likely that SSTs will return to pre-jump levels before September 2025. In the few simulations that do not simulate a return to pre-jump levels, SSTs revert to the expected warming trajectory over the following years. Thus, SSTs have not shifted to a higher or accelerated warming trajectory after a record-shattering SST jump in the models. The ability of climate models to simulate both the magnitude of the SST jump and the timing of the decline of positive SST anomalies enhances confidence in their use for future studies to understand the length, intensity and drivers of such extreme events, and to quantify their impact on regional weather systems and their potentially devastating consequences for terrestrial and marine ecosystems, and their services.

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

## Methods

### Observations

Observed SST anomalies were calculated using the global and highly resolved NOAA OISST V2.1[1] product, which is based on observations from satellites, ships, buoys and Argo floats. Among the different available SST products[45], we chose NOAA OISST V2.1[1] as it has shown the best performance[45]. In addition, NOAA OISST V2.1[1] is also the only dataset that includes the observations from the Argo programme that was started in 1999 and has operated between 3,000 and 4,000 Argo floats since 2007. Monthly NOAA OISST V2.1[1] observation-based estimates from 1982 to November 2024 were downloaded (last accessed on 4 December 2024). Although this dataset is used in the main paper, the underlying warming trend, the magnitude by which the record-shattering jump in SSTs in 2023–2024 broke the previous SST record, and the return period of such record-shattering jumps in SST were also quantified for comparison using the SST Analysis production version 3.0 from the European Space Agency Sea Surface Temperature Climate Change Initiative based on the Operational Sea Surface Temperature and Ice Analysis (OSTIA) reanalysis system ICDR3.0 (Integrated Climate Data Record 3.0)[46]. This dataset is also among the better-performing observation-based SST estimates with a high spatial resolution[45] and covers the years from 1980 to 2024. In addition, longer observation-based SST time series from one reanalysis product (ERA5[30]) and two interpolated observational products (HadISST[31] and ERSST[32]) from 1940 to 2023, the years that are covered by all three SST products, were used to estimate the variance and the autocorrelation of the natural temperature variability of annually and globally averaged SSTs. The autocorrelation and variance are both necessary to construct the synthetic time series that are used to estimate the return period of record-shattering SST jumps (see section 'Observation-based estimate of the return period of record-shattering SST jumps' below).

Here we chose to calculate annual averages starting in April as this is the month when SSTs started to break records by a larger margin (Fig. 1a). However, differences to previous records in NOAA OISST V2.1[1] are up to 0.009 °C larger when averaging from May to April, June to May, and July to June, and become smaller afterwards. The number of times that such events occur in climate models varies between 11 and 12 when the first month over which one calculates the annual average is between April and July. The results are hence similar for averaging periods starting later in the year.

### Observation-based estimate of the return period of record-shattering SST jumps

An observation-based estimate and the associated uncertainties of the return period of record-shattering SST events that break the previous record by the margin of 2023–2024 are constructed based on synthetic time series of lengths of 100 million years. These time series were obtained using the warming trend and an autoregressive model of first order (AR(1)), which relies on the temperature variance and the lag-one-year autocorrelation.

The underlying warming trend in SSTs in 2023–2024 was estimated using an Enting spline[47] that was fitted to the observation-based globally and annually averaged SSTs from 1982–1983 to 2023–2024. From this spline fit, the trend in 2023–2024 was calculated as the slope of that spline fit in 2023–2024. The Enting spline filters short-term variability from time series with noise. The amount of noise that is removed depends on the cut-off period, with low cut-off periods removing only the short-time variability and large cut-off periods removing both the short-term and longer-term variability. To determine the cut-off period that allows a robust estimate of the trend in 2023–2024, the trend in the 11 climate simulations that include a record-shattering jump in SSTs was calculated for cut-off periods ranging from 25 years to 55 years. Cut-off periods below 25 years do not filter the decadal variability, and cut-off periods above 55 years are too rigid to capture nonlinear

components[47] of long-term warming trends. In each simulation with a record-shattering jump, the Enting spline was fitted to the year of the jump and the 41 years before. The length of 42 years was chosen as NOAA OISST V2.1[1] also covers 42 years. The so-estimated trend was then compared with the 'true' underlying trend. This 'true' trend was estimated with a 31-year running mean that can be calculated for the jump years in models because the SSTs after the jump are also simulated by the models. An Enting spline with a cut-off period of 40 years fits that underlying 'true' trend best and is only 1 ± 18% larger, that is, 0.003 ± 0.049 °C per decade). With this approach applied to NOAA OISST V2.1[1], we estimated the trend in 2023–2024 to be 0.269 °C per decade. The uncertainty of the estimated trend was quantified using the synthetic time series of 100 million years with a trend of 0.269 °C per decade and the best estimate of the variance and autocorrelation (see paragraph below). At each of the around 225,000 record-shattering jumps in this time series, the trend was estimated with an Enting spline with a cut-off period of 40 years. The standard deviation of these estimated trends is 0.037 °C per decade. Thus, the estimated trend from the Enting spline has a likelihood of 95% to be between 0.195 and 0.343 °C per decade.

To further quantify the sensitivity of the resulting trend to the estimation method, we fitted a linear trend from 2004/2005 to 2023/2024 to approximate the trend in the year of the jump in NOAA OISST V2.1[1]. The slope of this linear fit is 0.254 °C per decade, slightly smaller than the estimate from the Enting spline (0.269 ± 0.037 °C per decade) as it does not capture the slight nonlinear warming component[48], but well within the uncertainty range. These sensitivity tests highlight that the largest uncertainty of the trend estimates results from the climate variability that is superimposed on the underlying trend and not from the method that is used to determine the trend.

As opposed to the estimate of the trend, we could not rely on observation-based SST data over the satellite period, such as NOAA OISST V2.1[1], for the estimation of autocorrelation and variance, as the relatively short length of the satellite period (1982 until present) makes uncertainties of the autocorrelation and variance large. Instead, the observation-based variance and lag-one autocorrelation are estimated from the three SST products described above over the period 1940–2023 after detrending the data using a cubic spline, resulting in three estimates each. As it is not evident which product performs best, the mean of all three estimates is used as the most likely estimate of the autocorrelation and variance. An implicit assumption for determining the uncertainty of autocorrelation and variance estimates is that the temperature variability follows a Gaussian distribution (Extended Data Fig. 6). For the ERA5[30], HadISST[31] and ERSST[32] products, three tests (Kolmogorov–Smirnov, Shapiro–Wilk and Anderson–Darling) were employed to test whether the SST anomalies follow a Gaussian distribution. Across these 3 tests and 3 time series, the P values vary from 0.26 to 0.90, indicating no significant deviation from a Gaussian distribution. We thus conclude that the SST data are well modelled by a Gaussian distribution.

To estimate the uncertainty of the autocorrelation and variance, sampling distributions for the best estimates of the variance and autocorrelation were constructed. For normally distributed time series of a given length, theoretical sampling distributions of variance and autocorrelation estimates are known[49,50]. These sampling distributions characterize the dispersion of estimates around a true value. The variance of the sampling distribution thus informs about the uncertainty in the estimate from internal climate variability for an estimate from a single product. As the best estimates of the variance and autocorrelation are both an average of three products, the uncertainty is smaller than the uncertainty from a single product. To reflect this reduced uncertainty, we randomly sampled 10,000 times 3 values from the respective distribution and averaged over these 3 values. The resulting values are then the sampling distribution of the average estimate of the three products. The resulting uncertainties are relatively large

(Extended Data Fig. 7b,c) as the internal climate variability still renders the estimation of the actual variability and autocorrelation uncertain in a time series that covers 84 years from 1940 to 2023. These large uncertainties owing to the internal variability cover the individual estimates from the three products (Extended Data Fig. 7). Furthermore, the sensitivity tests towards the detrending method were evaluated. When using a second-order polynomial or an Enting spline with a cut-off period of 40 years for detrending instead of a cubic spline, the results change by ±10% and are within the large uncertainties in the estimates resulting from the internal climate variability.

We then constructed 10,000 AR(1) models by randomly sampling 10,000 combinations of trends, autocorrelations and variances from the distributions of the respective quantities (see histograms in Extended Data Fig. 7). For each of these 10,000 AR(1) models, a synthetic time series of 100 million years was simulated to determine the return period of record-shattering SST jumps. Based on these 10,000 return period estimates, the distribution of the observational return periods was determined, with its spread representing the uncertainty in the return period estimate. The resulting mean and median of the return period are 512 years and 448 years, respectively, with a 95% confidence interval ranging from 205 years to 1,185 years. The lower bound of the confidence interval is the return period for which 2.5% of probability mass is distributed over lower return periods. Consistently, the upper bound is defined such that lower return periods have 97.5% of the probability mass.

To test the sensitivity to the choice of the observation-based SST estimate, the underlying warming trend, the magnitude by which the record-shattering jump in SSTs in 2023–2024 broke the previous SST record, and the return period of such record-shattering jumps in SST were also quantified using the SST Analysis production version 3.0 from the European Space Agency Sea Surface Temperature Climate Change Initiative based on the OSTIA reanalysis system ICDR3.0[46]. In that dataset, the SST jump in 2023–2024 is 0.23 °C, the underlying trend in 2023–2024 is 0.209 °C per decade, and the resulting return period estimate is 543 years with a 95% confidence interval of 204 years to 1,371 years. Thus, the return period is almost the same as the return period estimated based on NOAA OISST V2.1[1]. Furthermore, the number of record-shattering events in climate models for a jump of 0.23 °C is 19, resulting in a return period of 583 years. Thus, the estimates of the return period of record-shattering events based on observations and models are closer when using this dataset instead of NOAA OISST V2.1[1].

The sensitivity of the return period estimate to the choice of the underlying statistical model was also tested. In addition to using the AR(1) model, the return period was calculated with an autoregressive model of order 2 (AR(2) model) and a moving average of order 1 model (MA(1) model). The resulting mean return periods are 484 years (196–1,142 years, 95% confidence interval) for the AR(2) model and 498 years (201–1,179 years) for the MA(1) model. As the results are almost indistinguishable from the results with the AR(1) model that resulted in a return period of 512 (205–1,185) years, we here rely on the well-established AR(1) model.

In addition to these 10,000 time series, a smaller number of time series was constructed for a range of combinations of autocorrelations, standard deviations and trends to visualize the respective effect of the effect of each quantity on the resulting return period of record-shattering jumps in SSTs and to be able to compare the models with observation-based estimates in terms of autocorrelations, standard deviations, trends and return periods (Extended Data Fig. 2).

## Climate model simulations
We used 270 simulations from 35 coupled climate models: 170 simulations are from the Coupled Model Intercomparison Project Phase 6 (CMIP6; Extended Data Table 1), 30 simulations are from the GFDL-ESM2M[41] large ensemble from the University of Bern[42] (ensemble members are numbered 1 to 30), and 50 simulations are from the CESM2[43] large ensemble[44] (ensemble members 51 to 100). Although the CESM2-LE contains 100 ensemble members, only the second half of the members was used, as the first half of the members had too high temperature variability owing to too high sensitivities of aerosol–cloud interactions to variability in biomass burning[44].

The CMIP6 and CESM2-LE simulations were forced with historical data from CMIP6 until 2014 and with the Shared Socioeconomic Pathways (SSPs) 5-8.5 (CMIP6 simulations) and 3-7.0 (CESM2-LE)[51]. The GFDL-ESM2M-LE simulations were forced with historical data from CMIP5 until 2005 and with Representative Concentration Pathway (RCP) 8.5 afterwards[52]. The resulting radiative forcing between SSP3-7.0, SSP5-8.5 and RCP8.5 is smaller than 3% in 2020, smaller than 5% in 2030 and smaller than 7% in 2040[53,54], resulting in global warming that is statistically indistinguishable until 2040[29].

To compare the climate models with the observed record-shattering jumps in globally averaged SSTs over 2023–2024, annual SST means were calculated from April to March, the part of the year when the observed record-shattering jump occurred in 2023–2024. For each model, monthly data from each model's original grid was analysed to avoid introducing errors by regridding the data first.

The climate model return period for record-breaking global jumps in SST that exceed the previous record by at least 0.25 °C was estimated by counting the number of record-breaking jumps in SST simulations between 2000 and 2040, the years when the trend is approximately similar to the trend in 2023–2024. For the climate model return period estimate, a confidence interval for the return period estimate of record-breaking global jumps in SST was constructed by identifying the counted number of such events in the model simulations (here 11) with the outcome of a binomial experiment[55]. The binomial parameter $p$ represents the probability of any year in any model simulation to show such an event. We then calculated the 95% Pearson–Clopper confidence interval ($p_{lower}$, $p_{upper}$) for the binomial parameter $p$ (refs. 55,56). As the return period is given by $1/p$, the confidence interval of the return period is ($1/p_{upper}$, $1/p_{lower}$). Here, the confidence interval of the return period is 562–2,016 years. The Pearson–Clopper confidence interval was chosen over the confidence interval assuming a normal distribution for $p$, as $p$ is close to zero where normality cannot be assumed[55]. Nonetheless, a relatively similar confidence interval (633–2,459 years) would be estimated when assuming a normal distribution. This similar result suggests that the confidence interval is relatively insensitive to the chosen approach.

We have not detected an unusual proportion of record-shattering SST jumps in any of the model ensembles. Therefore, most record-shattering jumps in SSTs are found in model simulations that tend to have the largest number of ensemble members (2 such events in 30 GFDL-ESM2M-LE members, 2 in 50 CESM2-LE members, 2 in 50 CanESM5 members, 4 in 43 MIROC6 members, and 1 in 6 IPSL-CM6A-LR members). Given that the observation-based estimate of the return of record-shattering jumps in SSTs has a confidence interval between a 1-in-a-205-year event and a 1-in-a-1,185-year event, it appears plausible that no such events are sampled in model ensembles with 10 or fewer ensemble members, which cover a maximum of 410 years. The smallest ensemble in which an event was simulated was IPSL-CM6A-LR, which runs over 246 years. Finding such an event in one small ensemble is also plausible given that there are many such small ensembles without a record-shattering SST jump. Lastly, the most events were found in the MIROC6 ensemble with 4 events in 1,763 years (43 ensemble members). This corresponds to a return period of around 440 years, within the uncertainty range of the observation-based estimate.

The climate model simulations analysed here have temperature trends, temperature variabilities and autocorrelations that spread around the observation-based estimates of these three variables (Extended Data Fig. 2). To compare the trends, temperature variabilities and autocorrelations in the models with the observation-based estimates, we estimated the quantities in the same way as we did for

the observation-based estimates. The multi-model mean temperature trend in 2023–2024 in the models is 19% smaller (43% smaller to 9% larger, interquartile range) than the observation-based trend estimated from NOAA OISST V2.1[1] (0.27 °C per decade) in CMIP6 model simulations, 26% smaller (11–44% smaller) than the observation-based trend in CESM2 simulations, and 37% smaller (25–50% smaller) in GFDL-ESM2M simulations. The multi-model mean standard deviation of the temperature variability (temperature from 1940 to 2023 after being detrended with a third-order polynomial) is 16% higher (3% smaller to 40% larger) in CMIP6 model simulations than the average of the observation-based estimates based on ERA5[30] (0.085 °C), HadISST[31] (0.082 °C) and ERSST[32] (0.092 °C), and 15% (10–21%) and 20% (14–26%) higher than the observation-based estimate in CESM2 and GFDL-ESM2M simulations, in line with the overestimation of decadal trends of major climate modes in CMIP6 models[57]. In addition, the multi-model mean autocorrelation is 55% (31–86%) larger in CMIP6 models than the average of the observation-based estimates based on ERA5[30] (0.20), HadISST[31] (0.31) and ERSST[32] (0.41), 23% (8–38%) larger in CESM2 simulations, and 19% (8–37%) larger in GFDL-ESM2M simulations. A part of the difference between the models' and the observed parameter estimates may be due to the uncertainty in the observed estimates (standard deviations for the parameter estimates of the SST trend, SST standard deviation and year-to-year SST autocorrelation of 0.03 °C per decade, 0.004 °C and 0.06, respectively). In addition, the counting of the record-shattering jumps in globally averaged SSTs led to a return period of 1,000 years, higher than the observation-based estimate of the return period (512 years) but within the confidence interval (205 years to 1,185 years). The slightly higher simulated return period might well be the result of the slight tendency towards relatively high autocorrelations and relatively small trends, compensated by a relatively high standard deviation. However, the observation-based estimates of the return period might also just be affected by the internal variability leading to a too-low best guess based on observations.

### Regionally averaged time series

Time series of spatially averaged SST anomalies were calculated for four different regions: 60° S to 60° N, the El Niño 3.4 index region in the Eastern Tropical Pacific from 5° S to 5° N and from 170° W to 120° W, the North Atlantic as defined by all Atlantic open-ocean biomes as defined by ref. 58 excluding the Mediterranean Sea and limited by 0° N, and the North Pacific as defined by the North Pacific subpolar seasonally stratified biome and the subtropical seasonally stratified biome from ref. 58. The mask for the respective biomes were mapped on each individual native model grid.

### Data availability

The Earth system model output used in this study is available via the Earth System Grid Federation (https://esgf-data.dkrz.de/projects/cmip6-dkrz/). The large ensemble from CESM2 is available at https://www.earthsystemgrid.org/dataset/ucar.cgd.cesm2le.output.html. The monthly two-dimensional SST output from the GFDL-ESM2M large ensemble is available at https://www.seanoe.org/data/00897/100853/. The NOAA OISST V2.1[1] data are available at https://psl.noaa.gov/data/gridded/data.noaa.oisst.v2.highres.html. The SST Analysis production version 3.0 from the European Space Agency Sea Surface Temperature Climate Change Initiative based on the OSTIA reanalysis system ICDR3.0[31] is available at https://catalogue.ceda.ac.uk/uuid/4a9654136a7148e39b7feb56f8bb02d2/. The ERA5 SST data are available at https://cds.climate.copernicus.eu/datasets/reanalysis-era5-single-levels-monthly-means?tab=overview, the ERSST SSTs are available at https://www.ncei.noaa.gov/products/extended-reconstructed-sst, and the HadISST SSTs are available as https://www.metoffice.gov.uk/hadobs/hadisst/data/download.html. All maps were created using the Basemap tool in Python (https://matplotlib.org/basemap/stable/).

### Code availability

The code that was used for this study is available on Zenodo at https://doi.org/10.5281/zenodo.14618176 (ref. 59).

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

**Acknowledgements** J.T. was funded by the Swiss National Science Foundation under grant PZ00P2_209044 (ArcticECO). T.L.F. acknowledges funding from TipESM 'Exploring Tipping Points and Their Impacts Using Earth System Models', which is funded by the European Union. Grant agreement number: 101137673. F.A.B. and T.L.F. acknowledge funding from the Bloom Foundation. T.F.S. acknowledges support by the Swiss National Science Foundation (project 200492). We acknowledge the World Climate Research Programme, which, through its Working Group on Coupled Modelling, coordinated and promoted CMIP6; the climate modelling groups for producing and making available their model output; the Earth System Grid Federation (ESGF) for archiving the data and providing access; the multiple funding agencies who support CMIP6 and ESGF; the CESM2 Large Ensemble Community Project and supercomputing resources provided by the IBS Center for Climate Physics in South Korea; and the Swiss National Supercomputing Centre for providing resources for the GFDL-ESM2M large ensemble.

**Author contributions** This idea for the study was conceived by J.T. A detailed outline was developed by J.T. in collaboration with T.L.F. and T.F.S. J.T. performed the model output analysis and produced the figures. F.A.B. built the statistical models for the observations-based estimate of the return period of the jump in SSTs. The CMIP6 SST time series (60° S–60° N, El Niño 3.4 index, North Pacific and North Atlantic) were provided by L.V., and the time series for the same regions from the large ensembles of CESM2 and GFDL-ESM2M were provided by J.T. The GFDL-ESM2M-LE simulations were conducted by F.A.B. with guidance by T.L.F. The initial paper was written by JT. All authors contributed ideas, discussed the results and wrote the paper.

**Funding** Open access funding provided by University of Bern.

**Competing interests** The authors declare no competing interests.

**Additional information**
**Correspondence and requests for materials** should be addressed to Jens Terhaar.

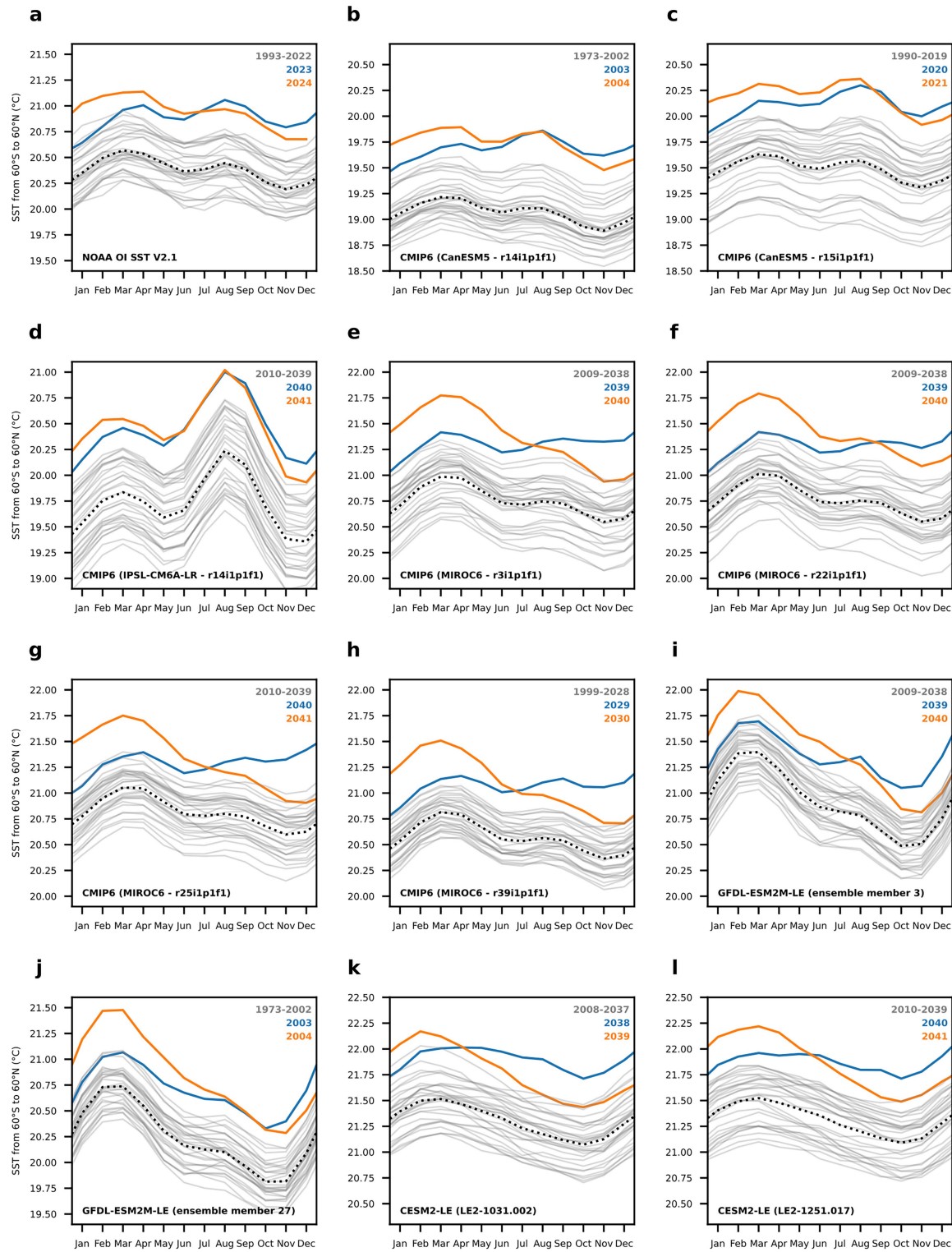

**Extended Data Fig. 1 | Record-shattering jumps in global sea surface temperature as observed in 2023/24 are simulated by climate models between 2000 and 2040.** Sea surface temperature (SST) climatologies for record-shattering (record-breaking events of largest magnitude before 2024) annual (April-March) global (60°S-60°N) SST events for **a)** observations from NOAA OISST V2.1[1], and climate model simulations from **b)-h)** CMIP6 (CanESM5[39,40] – r14i1p1f1, CanESM5[39,40] – r15i1p1f1, IPSL-CM6A-LR[60] - r14i1p1f1,

MIROC6[61] – r3i1p1f1, MIROC6[61] – r22i1p1f1, MIROC6[61] – r25i1p1f1, MIROC6[61] – r39i1p1f1), **i)-j)** the GFDL-ESM2M-LE[41,42] (ensemble members 3 and 27), and **k)** – **l)** the CESM2-LE[43,44] (ensemble members LE2-1301.002 and LE2-1251.017). The years of the onset of the respective events are shown as blue lines, the years of the decline as orange lines, and the 30 preceding years as grey lines with a black dotted line showing the average of these preceding 30 years.

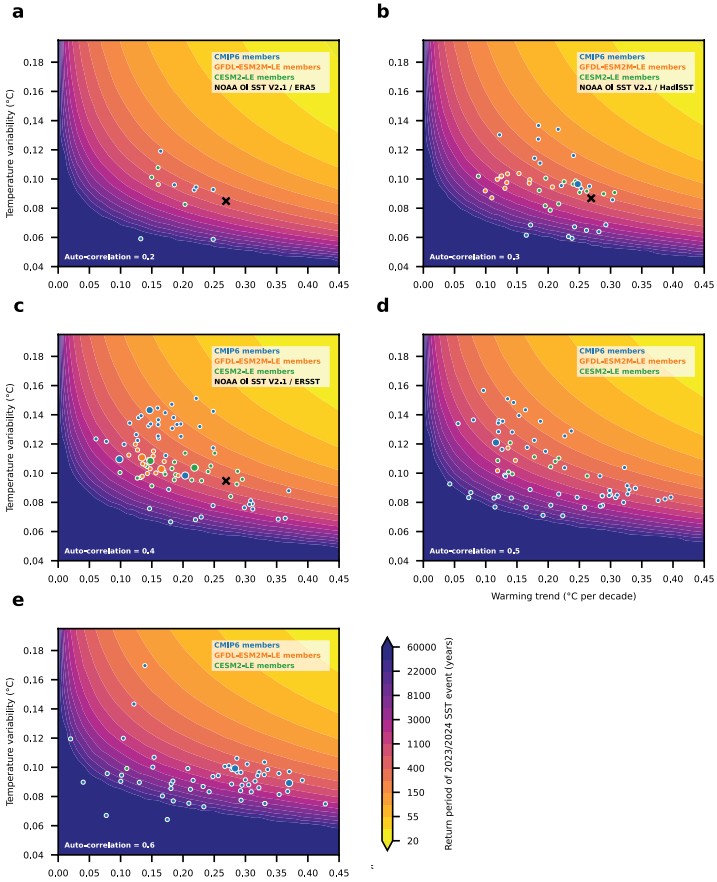

**Extended Data Fig. 2 | The return period of record-shattering jumps in sea surface temperature depends sensitively on the warming trend, and the autocorrelation and variance of annual sea surface temperatures.** The return period of record-shattering sea surface temperature (SST) events, defined as events when the annually averaged SST (April to March) between 60°S and 60°N is at least 0.25 °C (record-breaking SST observed in 2023/2024) larger than the previous record SST, is plotted in dependence of the warming trend over a given period and the temperature variability for five different autocorrelations: **a)** 0.2, **b)** 0.3, **c)** 0.4, **d)** 0.5, and **e)** 0.6. The return period is calculated based on timeseries of 100 million years with an AR(1) model and prescribed trends, variability and autocorrelation (see methods). The dots indicate each simulation, and their location indicates the trends, variability and autocorrelation derived from these simulations. The larger dots show the simulations that included a record-shattering jump in SSTs. Accordingly, the crosses indicate the trend from NOAA OISST V2.1[1] and variability and autocorrelation from ERA5[30] (cross in **b)**), HadISST[31] (cross in **c)**), and ERSST[32] (cross in **d)**).

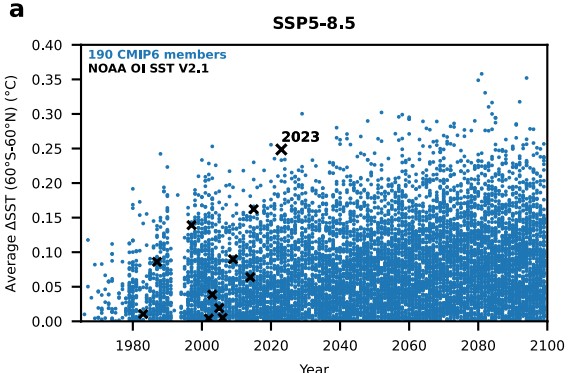

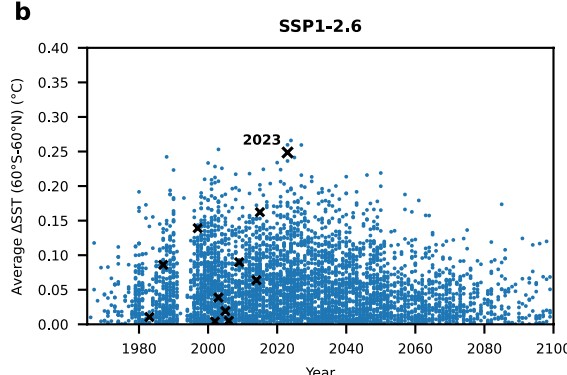

**Extended Data Fig. 3 | The return period of record-breaking annual sea surface temperature events over the 21st century depends sensitively on the emission scenario.** Record-breaking annual (April-March) global (60°S-60°N) sea surface temperature (SST) events based on 190 climate model simulations from CMIP6 (blue dots) (Extended Data Table 1) for **a)** the high-emission SSP5-8.5 and **b)** the low-emission SSP1-2.6. In addition, record-breaking SST events based on SST observation-based estimates from NOAA OISST V2.1[1] are shown as black crosses.

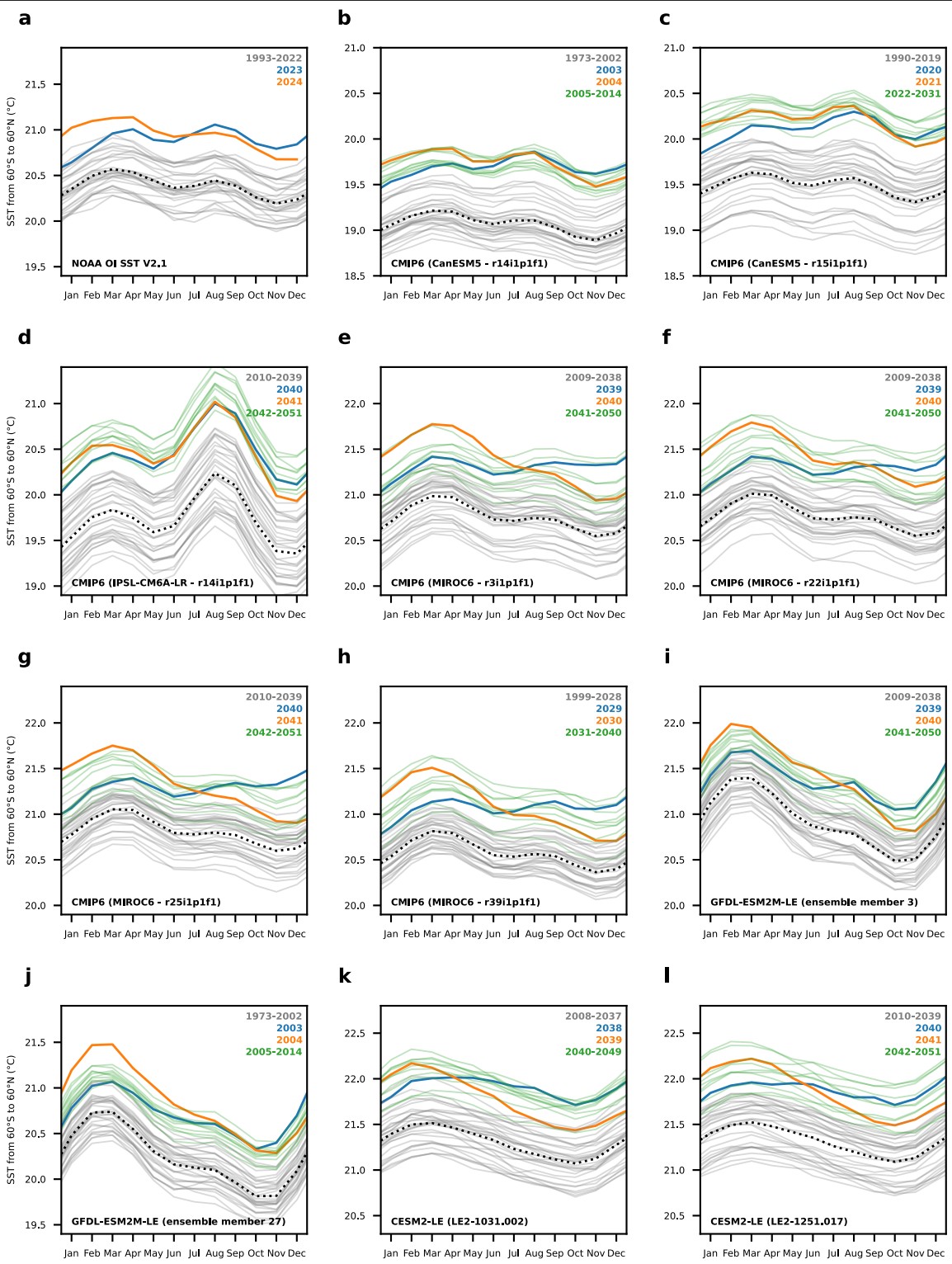

**Extended Data Fig. 4 | The gap in global sea surface temperatures created by the simulated record-shattering jumps is closed in most simulations in the years following the jump.** Sea surface temperature (SST) climatologies for record-shattering (record-breaking events of largest magnitude before 2024) annual (April-March) global (60°S-60°N) SST events for **a)** observations from NOAA OISST V2.1[1], and climate model simulations from **b)-h)** CMIP6 (CanESM5[39,40] – r14i1p1f1, CanESM5[39,40] – r15i1p1f1, IPSL-CM6A-LR[60] - r14i1p1f1, MIROC6[61] – r3i1p1f1, MIROC6[61] – r22i1p1f1, MIROC6[61] – r25i1p1f1,

MIROC6[61] – r39i1p1f1), **i)-j)** the GFDL-ESM2M-LE[41,42] (ensemble members 3 and 27), and **k) – l)** the CESM2-LE[43,44] (ensemble members LE2-1301.002 and LE2-1251.017) between 2000 and 2040. The years of the onset of the respective events are shown as blue lines, the years of the decline as orange lines, the 10 years following the event as green lines, and the 30 preceding years (pre-jump years) as grey lines with a black dotted line showing the average of these preceding 30 years.

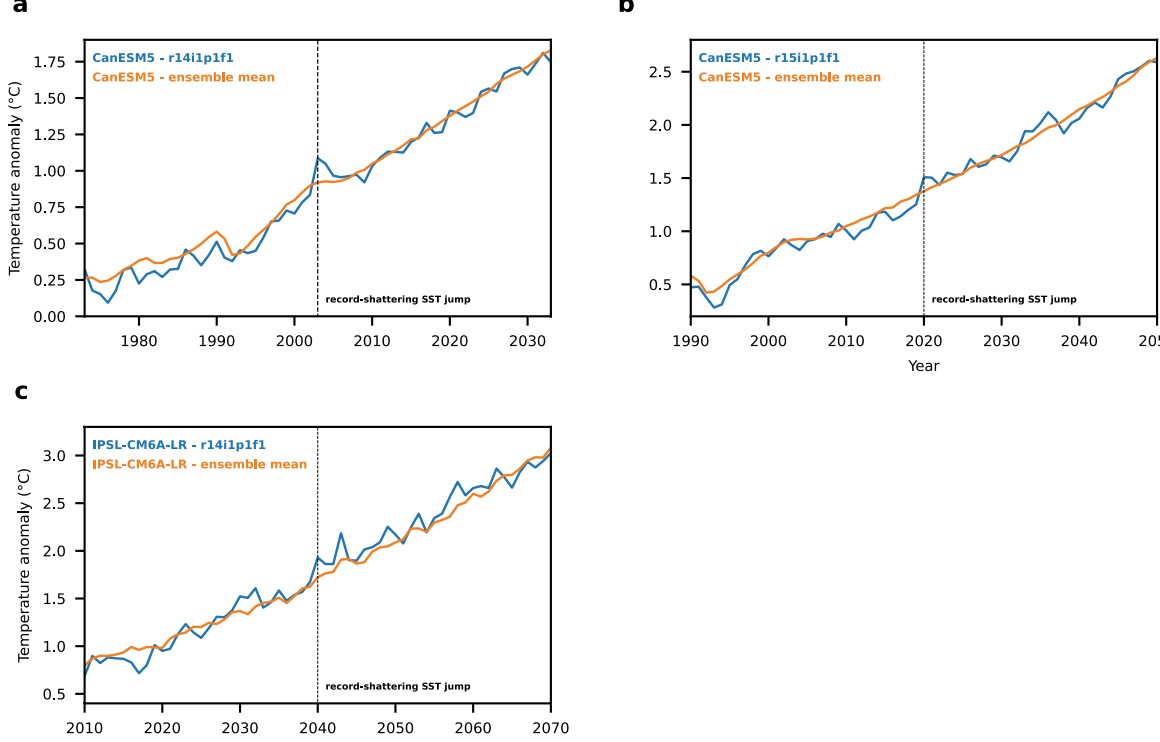

**Extended Data Fig. 5 | Annual global sea surface temperature anomalies return to the long-term warming trend after record-shattering jumps in sea surface temperatures that exceed previous records by more than 0.25 °C.** Annual (April-March) global (60°S-60°N) sea surface temperature (SST) for the 71 years around a record-shattering jump in SSTs that exceeds the previous record by at least 0.25 °C for the simulations **a)** CanESM5[39,40] – r14i1p1f1,

**b)** CanESM5[39,40] – r15i1p1f1, and **c)** IPSL-CM6A-LR[60] – r14i1p1f1, the simulations in which SSTs do not come back to pre-jump levels in the future (blue lines). The respective ensemble means for CanESM[39,40] (50 ensemble members) and IPSL-CM6A-LR[60] (6 ensemble members) are shown in comparison (orange lines). The year of the jump in SSTs is shown as a dashed black line.

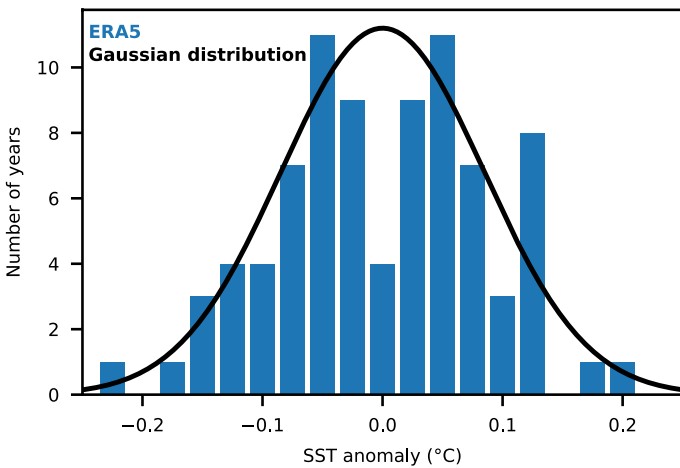

**Extended Data Fig. 6 | Observation-based sea surface temperature anomalies can be approximated by a gaussian distribution.** Distribution of global (60°-60°N) annual (April to March) Sea surface temperature (SST) anomalies for the observation-based reanalysis product ERA5[30] from 1940 to 2023 (blue line). After detrending (see Methods), the anomalies were binned in 21 bins of 0.025 °C each. In addition, a gaussian distribution with the standard deviation of the SST anomalies is shown.

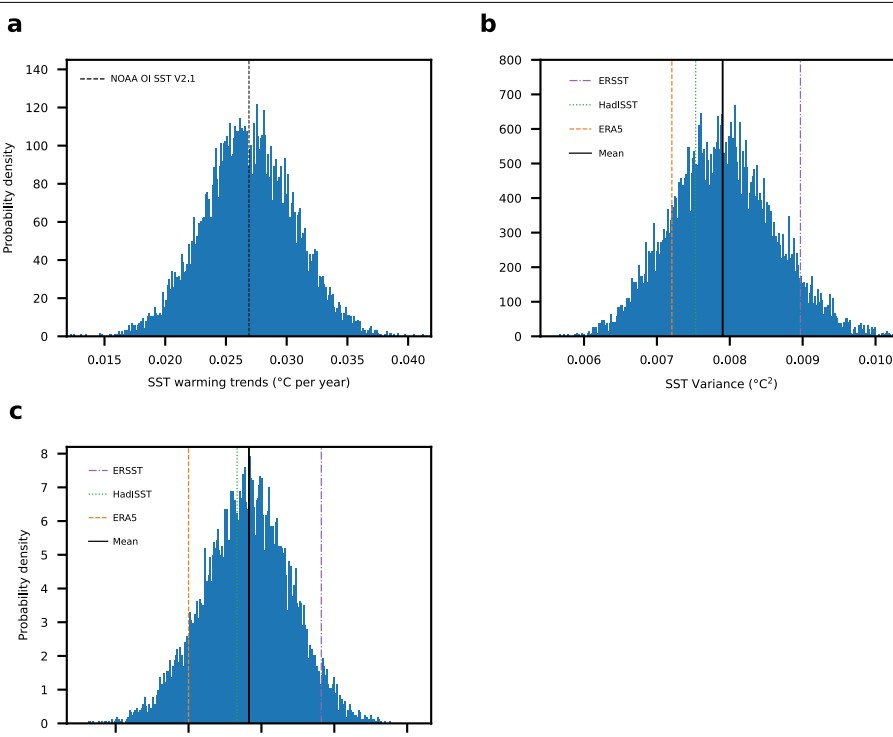

**Extended Data Fig. 7 | Sampling distributions for the 3-product averages of variance and autocorrelation and of the single trend estimate.** The single estimate for the trend is shown as a dashed line in **a)** and the three different estimates are shown as dashed orange (ERA5[30]), dotted green (HadISST[31]), and dash-dotted purple (ERSST[32]) lines in **b)** and **c)** together with the mean estimate of the three individual estimates as a solid black line.

**Extended Data Table 1 | List of CMIP6 climate models and the number of ensemble members per model used in this study**

| Climate model | Ensemble members | Reference |
|---|---|---|
| ACCESS-CM2 | 3 | [62] |
| ACCESS-ESM1-5 | 10 | [63] |
| BCC-CSM2-MR | 1 | [64] |
| CanESM5-CanOE | 3 | [39,40] |
| CanESM5 | 50 | [39,40] |
| CAS-ESM 2 | 2 | [65] |
| CESM2 | 3 | [43] |
| CESM2-WACCM | 1 | [43] |
| CIESM | 1 | [66] |
| CMCC-CM2-SR5 | 1 | [67] |
| CMCC-ESM2 | 1 | [68] |
| CNRM-CM6-1-HR | 1 | [69] |
| CNRM-CM6-1 | 6 | [69] |
| CNRM-ESM2-1 | 4 | [70] |
| EC-Earth3 | 2 | [71] |
| EC-Earth3-Veg-LR | 3 | [71] |
| EC-Earth3-Veg | 5 | [71] |
| FGOALS-f3-L | 3 | [72] |
| FGOALS-g3 | 4 | [73] |
| FIO-ESM-2-0 | 3 | [74] |
| GFDL-ESM4 | 1 | [75] |
| GISS-E2-1-G | 7 | [76] |
| HadGEM3-GC31-LL | 1 | [77] |
| HadGEM3-GC31-MM | 1 | [77] |
| INM-CM4-8 | 1 | [78] |
| INM-CM5-0 | 1 | [79] |
| IPSL-CM6A-LR | 6 | [60] |
| MCM-UA-1-0 | 1 | unknown |
| MIROC6 | 43 | [61] |
| MIROC-ES2L | 6 | [80] |
| MPI-ESM1-2-HR | 2 | [81] |
| MPI-ESM1-2-LR | 10 | [82] |
| MRI-ESM2-0 | 1 | [83] |
| NorESM2-LM | 1 | [84] |
| NorESM2-MM | 1 | [84] |

The following references are cited in this table: refs. 62–84.