## [Peer Review file · Nature]

Record sea surface temperature jump in 2023/24 unlikely but not unexpected

Corresponding Author: Dr Jens Terhaar

Version 1:

Reviewer comments:

Referee #1

(Remarks to the Author)

Review of Article "Likelihood of the record-shattering global marine heatwave of 2023 and 2024" by Terhaar et al., in review in Nature

General comments

This is an informative and robust analysis on a currently still on-going multi-year global marine heatwave event in 2023 and 2024. The results suggest that although the sea surface temperatures were "record-shattering" during this event, they are not found incompatible with climate model simulations considering human-induced climate change. The authors thus conclude that while the event could clearly not have happened without human-induced global warming and remains a very rare event in the present climate conditions (about 1-in-1000 year event), it is not a sign of unexpected, accelerated warming.

Overall, the results will certainly be useful to inform the research community on the likelihood of the recorded sea surface temperatures. The analysis is sound and state-of-the-art. While the applied methods are maybe not groundbreaking, the study is very timely and thus of clear interest for a broad readership.

There are a few minor shortcomings in the discussion of the results, as well as aspects that the authors could have considered in more detail. I would recommend that they should consider these points in a revised version of the manuscript:

a) The 2023 and 2024 temperature anomalies were not only extreme in terms of the SST anomalies, but also in terms of the land temperature anomalies. This paper only addresses the likelihood of the sea surface temperature anomalies, not that of the global mean temperature anomalies. Hence, it only addresses part of the picture. The authors could potentially pinpoint this in the article. It should also be mentioned that large anomalies in sea surface temperatures do not necessarily result in high land temperatures (e.g. Seneviratne et al. 2014, Nature Climate Change; Sillmann et al. 2014, ERL), hence it would not be substantiated to suggest that the land surface temperatures were the result of the sea surface temperatures. Possibly, the authors could also extend their study to assess the probability of the global mean temperature anomaly in the analysed climate models and whether their conclusions still hold.

b) The authors should be careful in some of their statements about the event not being the sign of unexpected, accelerated warming. Since the event is still on-going, if the sea surface temperature anomalies were to last much longer, this event would become at some point incompatible with the existing climate models. So, this conclusion is reasonable at the present point in time, but its long-term relevance will depend on the further evolution of the event in the coming 12-14 months. On the other hand, the article does mention what type of anomaly would no longer be consistent with climate models (an anomaly that would last longer than another 12 months) and this information will also be useful for the climate research community.

While these aspects could deserve more consideration in the article, this is a robust and helpful study for the research community, which could be considered for publication subject to minor revisions taking into account the above comments.

Detailed comments

1) Abstract, lines 23-24: “and fueled weather extremes over land, for example causing extreme rainfall and droughts”

- I would remove this statement since the authors do not seem to provide any evidence for this statement in the text and analyses. If anything, past events such as the so-called “global warming hiatus” have rather shown a decoupling between sea surface temperatures and hot extremes over land (Seneviratne et al. 2014, Sillmann et al. 2014).

2) Abstract, lines 36-37: “Thus, the global MHW in 2023 and 2024 is likely not a sign...”. Add “so far” before “not a sign”. The authors highlight that this event a) is still on-going, and b) would be expected to return to pre-MHW levels between September of the 2nd year of the MHW and September of the following year; hence if the MHW continues beyond September of next year, the conclusion would be that this is actually an exceptional event that cannot be explained alone with the mean forcing so far.

3) Abstract, lines 37-38: “but rather a very rare event”; add at the end of this sentence “that was made more probable because of human-induced global warming”

4) Page 3, lines 50-51: “record-shattering is here defined as record-breaking global annual MHWs that exceed previous records by at least 0.25°C”: Why this threshold of 0.25°C? Can this be justified with statistics?

5) Page 9, line 159: “that coincidence”: the correct wording is likely “that coincide” here.

6) Page 15, line 260: “was likely a 1-in-a-1000-year extreme event and not a sign of unexpected, accelerated warming”: This wording could be misunderstood to mean that the MHW in 2023-2024 could have been induced by natural climate variability alone. However, the authors conclude clearly that this event can “entirely attributed to anthropogenic climate change”. It is important to state that the event was a 1-in-a-1000-year extreme event in the context of the human-induced climate change so far, but does necessarily go beyond the expectations associated with the present climate conditions in relevant climate simulations. Maybe change the wording to “was likely a 1-in-a-1000-year extreme event in the context of human-induced climate change to date, which could not have happened without human influence, but is nonetheless at this stage not a sign of unexpected, accelerated warming.”.

7) Page 15, lines 261-264: “As climate models... they can be used...”. While the authors have shown that the event so far is not incompatible with existing climate simulations, it was nonetheless extremely rare and is additionally still on-going. If it were to last much longer, there would be a point at which it would no longer be consistent with climate model simulations. The authors should be a bit more careful in formulating this conclusion.

References:

Seneviratne, SI, MG Donat, B. Müller, and LV Alexander, 2014: No pause in the increase of hot temperature extremes. *Nature Climate Change*, 4, 161-163

Sillmann, J, MG Donat, JC Fyfe and FW Zwiers, 2014: Observed and simulated temperature extremes during the recent warming hiatus. *Env. Res. Letters*, 9, 064023.

Referee #2

(Remarks to the Author)

Likelihood of the record-shattering global marine heatwave of 2023 and 2024

Jens Terhaar, Linus Vogt, Thomas L. Frölicher, Friedrich A. Burger, Thomas F. Stocker

Motivated by the unprecedented SST temperatures of 2023-24, this paper examines the likelihood of a warm global temperature record that exceeds the previous record by a large margin. The degree of exceedance from the previous record was unprecedented in the observational record. The study goes on to quantify the likelihood of such a high exceedance in various climate models

This question is of broad general interest, in particular whether such an event is consistent with our understanding of internal variability and whether climate models are able to capture such event.

I do have a number of general concerns however:

First is the writing in the document. I found it very hard to follow what the authors were doing in many parts of the manuscript, in particular I feel that the description of the analysis could be significantly improved. The reader loses track of the fact that you are focussing on record events with unprecedented increment in SST, not just really big events. I also find it unhelpful to call these events MHWs – you are examining a global and annual average, there’s no reason to expect that all regions experience MHW conditions for all or part of the year. To help with the narrative of the manuscript, it might be helpful to set out your questions up front e.g.: how likely is this kind of event? Would it be possible without anthropogenic warming? Are models able to simulate similar events? What can models tell us about the causes of such events? Could this event be the start of a regime shift to permanently warmer temperatures? Are such events likely to reoccur in the future? This way you have a clear motivation for the particular analysis that you carry out. The only question you articulate at the moment “whether this still ongoing event indicates an unexpected acceleration of global warming or simply a very rare extreme event” isn’t actually answered by your study.

It seems that the primary goal of this study is to establish the return period of an event like the one that occurred in 2023/24.

Given the significant biases in the ability of models to realistically simulate aspects of internal variability (something that is discussed in the literature, but hasn't really been discussed in this manuscript), I'm not sure that the proposed method is the best. For example you could simply construct a long synthetic timeseries based on observed temperature and establish the return period. You could then usefully use the large ensembles to estimate uncertainty bounds in the estimate of observed variability.

Moreover, using white noise to simulate the variability in annual temperatures is possibly over simplistic. The analysis around extended figure 2, for example indicates memory in temperature from one year to the next. The distribution of annual temperatures may also not be gaussian (was this tested?). This requires more careful consideration (e.g. you might consider an AR model). At least some sensitivity analysis should be made to your choices.

It would have been good to see some speculation, guided by the models, as to why this very rare event occurred. You showed for example the importance of an El Nino (but this isn't surprising as record warm events are typically associated with El Nino). But what causes such a large deviation from previous records. The models should be able to help answer this question.

Detailed comments:

33: In climate models, sea surface temperatures ...

Hard to understand what you mean here without more information. I suspect this will become clear after reading the manuscript, but its not from the abstract alone

44. resulting in a global marine heatwave ...

Just because the global average reached record temperatures does not imply MHW conditions globally (i.e. at all locations). I don't feel that this terminology is accurate

51: has been accompanied by atmospheric record warming > was accompanied by record atmospheric warming

78: The record-shattering global MHW in 2023 and 2024 has raised the question if this global MHW is a very rare extreme event or if climate change and global warming have been accelerating faster than expected based on fully coupled climate model simulations

We already know that this a rare event (its "record shattering"). And a rare event can be caused by anthropogenic warming, so the either/or doesn't make sense.

"climate change and global warming" whats the difference?

I don't see why you need "based on fully coupled climate model simulations" – you are interested in these questions irrespective of whether its consistent with climate models. This framing needs some work. How about: This raises questions around whether this event can be explained by natural variability alone, whether and how much it is affected by anthropogenic warming. And whether the extremeness of this event is unexpected even given the background warming that has occurred, based on comparison with climate models. If this is indeed the case it would suggest important shortcomings in the ability of our models to ...

79: accelerating faster than expected

You are describing a single event, I don't see that you can look at acceleration of warming rates based on a single year. I think this should be framed in terms of how exceptional the event was/is such an extreme event consistent with our models? Are the models missing something? NB even if this scale of event is unprecedented in models, that doesn't imply global warming is accelerating. Its more likely that the models are underestimating the degree of internal variability – something that has been illustrated in many prior studies

If the goal is to see if the observations are accelerating faster than climate models, why wouldn't you just compare the acceleration of the warming in model and observations. Why go to the trouble of looking at differences in return periods of extreme events?

82: be in uncharted territory ...

The event was unprecedented in the last 2000, so its already uncharted territory.

83: you don't need the 'so far'. Projections are the output of climate models

114 global (60°S-60°N)

This is not global

Figure 2. You haven't described what the figure is showing. My guess is that it's the temperature difference relative to the next highest record temperature, but this is not clear from any of the description.

125: Record-breaking heatwaves that are at least as large in magnitude as the record-shattering MHW over the last year were simulated 11 times between 2000 and 2040

This description is very confusing. If I understand correctly, you are not examining the magnitude of the event you are examining the difference in magnitude from the previous record. Maybe, you could call it something like a 'record increment' so its clear what you mean

127: surrounding the year

This would be 2003-2043

131: making last year's record-shattering MHW a 1-in-1000-year event

This is if we believe the climate models. To do this we would want to see that the models have a trend and variability that is consistent with the observations.

134: 74 times

Makes more sense to talk about a percentage

136-141: The climate models that simulate most of these record-shattering global MHWs ($>0.25^{\circ}\text{C}$) are mostly also the ones with the most ensemble members

This isn't really an interesting result. If record breaking events are evenly distributed, then we would expect models with more ensembles to have more records. More interesting is whether the proportion of records per ensemble is particularly unusual in certain models

Figure 3. The caption could be written more clearly. You are simply showing the April-march average temperature anomaly, relative to the preceding 30 years ...

146-148: what evidence do you have for these statements?

150: all record-shattering MHWs coincide with a positive El-Niño phase

Do you mean all the 11 events in the climate model simulations are associated with El Niño?

151: 2015/16, and 1997/98

So these years had the largest increments from the previous records?

160: did not break the old record by the same margin ...

This is the kind of terminology that you should be using throughout. Its far less ambiguous than the terminology used elsewhere.

162-175: As global and North Atlantic record-shattering MHWs are extremely unlikely and global record-shattering MHW patterns can vary in magnitude, an exceptionally large global MHW that occurs at the same time as an even less likely record-shattering MHW in the North Atlantic is even more unlikely.

I don't really see what point you are trying to make. High SSTA in the north Atlantic would make it more likely that the global average SSTA will be higher. But the NA only makes up a small part of the global ocean so this result is completely expected. Im not sure what value it adds to this study. Also if you are talking about the NA why not also the north Pacific or eastern tropical Pacific too?

191: 13 to 18 months after the beginning of the MHW...

This suggests that the ocean has a memory that exceeds 1 year timescales. This would suggest that your assumption of independent years for your synthetic timeseries does not really hold.

194: Only in 3 of the 11 record-shattering global MHWs, SST anomalies do not return to pre-MHW levels

In all the models there are some green lines that dip below the orange line. So this statement isn't completely accurate. Also you are only looking at the next 10 years, so your statement only applies to the following decade.

198: as indicated by the fact that the distances between the SSTs over the years ...

If there were say a 10% increase in the background warming rate, do you think you would be able to spot it just by looking at the distance between lines? Im skeptical.

201: The 3 cases where the gap is not filled again are simulated by CanESM5 and CESM2 ...

Which figure panels are you referring to? The clearest case for hot temperatures after the extreme year looks to be the IPSL model (panel d). CanESM2 has 2 ensembles one where it stays relatively hot, the other where it doesn't. Also looking at fig 4, it looks like the CESM2 ensemble has a systematically weak trend

My guess would be that its more about the strength of internal variability in the model rather than the warming trend.

Figure 4.

Its important to remember that there will be a large spread in the observational estimate, because of the length of the timeseries used to calculate the trend or the standard deviation. You can see this from the large ensembles. A single model (like the CESM2) will have the same forced trend and variability, but the estimated trend and variability differ because you are only using a few years to calculate these metrics. For example from the CESM ensemble Id guess that there's an error of about $\pm 0.1^{\circ}\text{C}/\text{decade}$ in the trend. (Indeed this might be a helpful to put error bounds on the observational return period)

229: best represent

Best compared to what? All other models? This wouldn't be supported by your analysis as there are blue circles from individual models that are closer to the observational estimate.

If you are comparing to the MMM I suspect that the CMIP6 MMM trend is closer to the observational estimate than the ESM2M model. Indeed the spread in the ESM3M model does not even overlap with the observations, which suggests to me that this wouldn't be a good representation of the real world. The reason that the return period isn't too bad is because of compensating errors – the trend is too weak, and the variability is too high.

260: was very likely a 1-in-a-1000-year extreme event

This model based estimate, is based on models that are clearly biased (based on your fig 4). Would a more reliable estimate be obtained using just the observational data to construct a long synthetic timeseries? The large ensembles could then be used to provide an estimate of uncertainty on the estimated variability. Assuming that variability is stationary at different levels of warming, the return rate would then depend on the rate of global warming.

231: The slight overestimation of the return period

I don't really see an overestimation. If you draw a line of constant return period through the observations, there seems to be almost equal numbers of models above and below.

235: only the models with high transient climate responses, such as IPSL-CM6A-LR, CESM2, and CanESM5

As above, for the two large ensembles, your analysis of fig. 4 points to an overly weak trend not an overly strong trend, over the decades considered)

343: Im not sure that ERA5 SST is the best choice as it can contains inhomogeneities in time. Products like HasISST or ERSST are designed for looking at temperatures over long timescales

352: You should indicate the number of different models

367-368: Im not clear why you would even talk about re-gridding as you are dealing with global or regional averages

379 onwards: I found it difficult to follow what you are doing here. The description needs significant improvement. I think a large part of the problem is that you don't say what you are calculating the return period of. You are looking for events where the globally averaged 2023 April-2024 March is the hottest on record and the record exceeds the previous record by at least 0.25oC.

380-390 I don't find this very clear. I think that you are constructing synthetic timeseries with different standard deviation and trend to look at how often a new record temperature exceeds the previous record by at least 0.25C

387: Your extended figure 5 looks very bimodal, not gaussian. There are quantitative metrics to determine how close distributions are to being gaussian, rather than just looking at this by eye.

389: Have you looked at the temporal autocorrelation to check how independent consecutive years are. I suspect that your results are going to be sensitive to the assumptions you make about the theoretical timeseries. I suspect some form of autoregressive model would be more appropriate.

391: The theoretical return period for observations and climate models

Not sure what you mean by this. Do you mean that you calculate the return period from a long synthetic timeseries with the same 2004-2023 trend and 1940-2023 standard deviation as the observations?

Id be sceptical of using a 20 year trend as a proxy for anthropogenic warming, if that is what you are aiming to do.

396: by subtracting a 3rd order polynomial

Why? Are your results sensitive your assumed shape of anthropogenic warming. E.g. do your results change if you use a linear fit or 2nd order polynomial

Table 1 & 2 could be summarised, just give the number of ensembles per model. (If the idea was to make the results reproducible you would also need to include simulation version numbers/download dates etc.)

398: 321 years for the CMIP6 ensemble, 596 years for the GFDL-ESM2M-LE, and 377 years for the CESM2-LE

If the observed return period is 440. Then 321 is not 30% of 440 and 596 is more than 100% of 440, so Im not sure what these numbers mean.

407: Again its hard to understand what you are doing.

405: In what way is 440 qualitatively similar to 321 or 596?

Version 2:

Reviewer comments:

Referee #1

(Remarks to the Author)

The authors have satisfactorily addressed my comments.

Referee #2

(Remarks to the Author)

This is a re-review of Likelihood of the record shattering jump in surface ocean temperatures in 2023 and 2024

First off Id like to thank the authors for diligently responding to all my comments. I think the paper is much improved. I do still have some comments/concerns that I feel need addressing. Ive also made a number of text modifications to help with

readability.

20: exceeding previous records

I think you mean: 'exceeding the previous record'

Also it would be worth stating when that record was set.

25: by 0.25°C

presumably "by at least 0.25°C"

27: been practically impossible and can thus be entirely

if its 'practically impossible' then its 'almost entirely' not 'entirely'!

Also I don't agree with this attribution statement. If its entirely related to AGW that means that it would have happened even without the extremely rare internally generated jump in temperature. I think its accurate to say that its practically impossible that this event would have happened without AGW, but the attribution statement is incorrect.

33: was so far an extreme event

Not sure what you mean with 'so far'? Sentence is confusing

34: 'return to'

I would suggest 'revert to'

38 This jump in SSTs ...

Need to delete 'jump in', or you are saying the same thing twice. i.e you want to say that both the sst and the jump in sst over the previous record is record breaking

41: from 2015/2016

... that occurred in 2015/2016

43: months of a year

months of the year

44: record-breaking jump in annual globally averaged SSTs

Is this definition specifically for the April to March period? If not how do you define this increment? Do you for example examine every 12 month (Jan-Dec, Feb-Jan, Mar-feb ...) period and compare it with all other equivalent 12 month periods?

46: , the record-shattering SSTs

You are talking about record shattering jump in SST, not record shattering SST. Its important to maintain the correct terminology

52: have stopped to be

are no longer

Figure 1

Given you are highlighting events that beat previous records, it might be helpful to also highlight the previous record year. If you colour the lines by year, then you could also demonstrate the gradual warming over time

69: Strong increases

Large increases

77 can occur

'can occur' is rather vague. If I take a random timeseries, record breaking jumps will always occur if you wait long enough (even without a warming trend)

It might be useful to list the prior record breaking jumps in order of magnitude (i.e. information from extended fig 3). In that way we can clearly see how exceptional 23/24 was

81: and whether this jump could have been expected based on fully coupled climate model simulations

This doesnt make sense as written, I think you mean ... and whether jumps of this size are simulated in climate models.

82: If state-of-the-art fully coupled climate models were not able to reproduce such

How about: The failure of state of the art climate models to reproduce ...

85: fully coupled

Delete? – no need to keep repeating this

85: were > are

86: simulate SUCH record-shattering

86: as the one observed over the last year
Delete?

87: and to project how SSTs continue to evolve after such events.
How about something like: ... to see if temperatures decrease again or remain high suggestive of tipping points in the climate system

93: output from 270 climate model simulations
output from 270 simulations spanning X different climate model

103: using an autoregressive model of order one (AR1)
I appreciate the effort the authors have gone to to include a more realistic model for SST variability. However, its still possible that the results here are sensitive to the model used. e.g. what happens if you used AR(2), ARIMA, ARFIMA ... models, each of which are based on certain assumptions about the original data? Some sensitivity testing, or at least some discussion around this topic is warranted.

107: under the current long-term warming trend
This relatively short period that is used to calculate the warming trend, will be sensitive to the internal variability (or even external factors like volcanoes). Doesnt this present a important uncertainty?
Using your large ensembles you can actually estimate the forced warming (using the ensemble average) and examine the degree of uncertainty associated with individual ensemble member. However you cant do this with the observations. Im not sure how best to handle this, but it should be given some careful consideration

110: Without human-made global warming, however, a record-shattering jump in globally averaged SSTs as observed last year would not have been possible
I think the statement is not quite correct. The trend that you are imposing isnt all related to anthropogenic forcing. I think it would be more correct to say without a strong long term warming, the chance of a record-shattering jump becomes negligible. Irrespective of the variability or autocorrelation characteristics, without a trend we found no record-shattering jumps in our synthetic timeseries.

119: years between 2000 and 2040
delete - you have just said this

123: 95% confidence interval; see Methods
How do you calculate the confidence interval in the models. You only have one set of 11070 years from which you count the number of record jumps

123: Even jumps in globally averaged SSTs that break the
124 previous record by a larger magnitude than 0.25C are not impossible in climate model simulations but increasingly unlikely (Fig. 3b).
How about: There are a small number of jumps that exceed 0.25, but the probability of a certain jump magnitude reduces with magnitude. For example jumps that exceed 0.20C are 6-7 time more likely than jumps exceeding 0.250C.

128-131: How about: Under high emissions scenarios the probability of record shattering events increases, in line with higher rates of warming. However under strong mitigation scenarios (S3A), the warming trend reduces in the future and there are no simulated record shattering events simulated across the model ensemble (S3B)

139: not only
This is not a proper sentence. I think you just need to delete 'not only' .

139: 'hot models'
Have you done this analysis taking into account the fact that different models have different numbers of years? i.e. your statement should be something like: climate models that have the highest rate of record shattering events at the hot models

140: Models with
However model with ...

143: In general, such extreme events are simulated by models that provide large ensembles
I dont understand this. Of course models with lots more years are much more likely to have more events. This doesnt tell us anything useful. You need to make like for like comparisons. If you wanted to do this you could subsample so that all models have the same number of years, or you could create long synthetic simulations based on each model.

168: the record-shattering jump in SSTs
Its confusing saying this here as there was no jump in 2023/24 on this region. Just say SSTA was once larger than in 2023/34. The same applies to the eastern tropical pacific below

169: In climate models, events as large as that in 2023/24 all coincide with an El-Ni.o event, i.e., a positive El Ni.o-Southern Oscillation phase of at least 1.5Å°C in the El-Ni.o 3.4 index (Fig. 4b-d).
Fig 4 only shows 3 model events. Are you saying this is consistent across all such events?

189: the simulated 11
the 11 simulated...

193: is 0.09°C smaller than the extraordinarily large observed 193 margin of 0.42°C
unnecessary to give both the 0.09 and the 0.42. One of these numbers will do

194 large regional
delete regional - unneeded as you say North Atlantic

194: in the 270 climate model simulations
Its more useful to talk about the number of years (11,000) than the number of simulations

197: An even larger model ensemble would be necessary to estimate the likelihood of the combination of a global record-shattering jump in SSTs and a record-shattering jump in SSTs in the North Atlantic in 2023 and 2024.
Not really. You could use exactly the same approach as you have done for the observations. Also I'm not really sure why we specifically care about this number, unless you are suggesting some specific mechanism that links global to NA SST

202-206: Repetitive
I find this NA analysis rather long and not very well motivated. Unless you could say for example that without the large jump in the NA then 2023/24 wouldn't have been a particularly noteworthy global event. For example you might show the temperature anomaly relative to 2016/17 to show which areas contributed most to the record shattering increase.

208: SST anomalies develop
SST typically anomalies develop

209: delete 'such'

210: stop to be
...stop being

212: have also stopped to be record breaking in
dropped below record levels

214: to their level before the 214 jump
return to their level before the 214 jump

216: In only
However ...

217: do not return to pre-jump levels
I can see this is true in panels c) and d) as the subsequent 10 years are all warmer than pre jump years. Is the other one f)?
You should explicitly explain what you mean by this, i.e. monthly global SST anomalies remain above all the same months in pre-jump years for the subsequent 10 years
Also its helpful to refer to specific panels in your figure

218: simulations. However, even in these three cases, the warming trend after the jump in SSTs has not accelerated as these models' SSTs warm less fast in the years following the jump and return to the respective ensemble mean within 8 years after the event (Extended Data Figure 5).
I find the statement confusing. I think you are saying is that after the event temperatures revert back to the long term trans over the course of a few year i.e. there's no shift to a new warming trajectory

224: significantly
Only use the word significantly if you mean that the difference is statistically significant.

224: rate32, facilitating continuing high levels of SST anomalies after the record-shattering jump in SST
Its ocean temperatures that will primarily set air temperatures over the ocean not the other way around

225: As SST anomalies only do not return to pre-jump levels in these 'hot models'³⁰, it is likely
Given that a return to pre-jump temperatures only fails to occur in these hot models ...

227: to where they were
temperatures mote typical of those...

227: in globally averaged SSTs
Delete

228-234: Hard to follow. I suggest you stick to what your analysis is actually showing you. I think the main thing you need to

say is that... Only under much larger warming rates than observed [and I suspect lower internal variability] might we expect temperatures to consistently remain above pre-jump years. Moreover, in the coming months we can expect to see temperatures revert back towards the long-term trend

237: a 1-in-a-760-year

I don't think providing a single number without uncertainty bounds is useful. Maybe just say 'a very rare'

239-242: This sentence is confusing. Break into 2 or 3 sentences. Also it's only some models that simulate these events, but they are very rare. And they are more common in models that have faster rates of anthropogenic warming

242: Furthermore,
Furthermore, in these models, ...

243: stop to be record breaking
drop below record levels

245: have stopped to be
Stopped being

248: warm slower over the following decade and return to the expected warming trajectory.
I find this statement ambiguous. This seems to imply the AGW slows down for some years. It's simply that temperatures revert to the long term mean warming trend over the span of a few years, as would be expected if this event was just related to internal variability (albeit a very rare jump)

248-253: I'm not sure it gives me great confidence - you only find these events in a subset of models. What provides confidence is if the model can simulate accurate warming rates and has variability that matches observations across all timescales. You wouldn't validate a model based on a single extremely rare event

349: within +/-10%.

In S7C you have one autocorrelation at 0.2 and the other at ~0.4. This is not a <10% difference. The same is true for the extreme values of the SST variance 0.0067 and 0.0083 are more than 10% different, so I'm not clear what you mean here

351: Extended Data Figure 7

I don't understand what is being presented in this figure. You have 3 data products and 3 detrending methods, where does the probability distribution come from?

In S7A what does the trend value represent - a 3rd order polynomial trend isn't represented by a single number?

Why are you using a single product in panel A and 3 products for the other parameters?

In C why can we only see lines for 2 data products.

Using colours rather than different linestyles would be much clearer

374: were calculated from April to March

I'm not really sure about the use of April to March. In the climate models you might find larger increments in other months. Also is the April - March jump actually the largest in the observations e.g. compared to May-April, Jun-May etc?

377: Most record-shattering jumps in SSTs are simulated by models that tend to have the largest number of ensemble members

Isn't this a trivial statement? More years so more events. Some account should be made for the overall length of data from different models

387: models is 20% smaller

I presume you mean the multi-model mean trend? Always more useful to provide a range (e.g. interquartile range)

387: 2004 to 2023

I don't really understand the justification for using a (linear?) trend from 2004-2023. This will be highly contaminated by internal climate variability. For example if you take one of your large ensembles and look at the differences in trends between members for 2004-2023 the differences will likely be quite large. Alternatively, if you just calculate the confidence interval in the trend it would be quite large. I think for a mean of 0.25°C/decade the uncertainty would probably be from 0.2 to 0.3 °C/decade

395: addition, the autocorrelation in CMIP6 models is 55%

you are presumably referring to the multi-model mean. This makes it sound like all models are 55% higher. These descriptions need to be more precise.

401-408: This information should go after describing the idealised timeseries

428: the Kolmogorov Smirnov

Shapiro-Wilk or Anderson-Darling tests are much more typically used to test for data normality, why have you used a KS test?

436: observational trend is estimated over the period 2004-2023 using the NOAA OI SST V2.11
Why three estimates of variance and autocorrelation and only one estimate of trend?

432-226: I find this description rather hard to follow. I think you could explain this more clearly, starting with your goal. E.g. In order to obtain an estimate of the uncertainty in the observational return frequency estimate of record shattering events (i.e. $>0.25^{\circ}\text{C}$ jump), we randomly sample trend, variance and autocorrelation estimates based on ... 10,000 times. 10,000 AR1 models are subsequently constructed and the return period is calculated.

Version 3:

Reviewer comments:

Referee #2

(Remarks to the Author)

This is a re-review of Likelihood of record-shattering jump in sea surface temperatures in 2023/24

Again I thank the authors for diligently responding to my new set of questions and comments. I have only minor suggested edits below.

One final general point is that there are now at least two studies I have seen describing some of the factors that attempt to explain the global or regional SST jumps in 2023:

<https://www.science.org/doi/10.1126/science.adq7280>

<https://www.nature.com/articles/s43247-024-01413-8>

It might be worth including a short description of the proposed mechanisms in the manuscript.

Regards

Alex Sen Gupta

18: Global ocean surface temperatures have been at record levels ...

Since about July 2024 they were no longer at a record level. So 'have been' should probably now be 'were'

22: of this still ongoing

As above this is no longer the case (sorry that's probably because of my long reviews!)

34: Since April 2023, global ocean (60°S-60°N; excluding cloudy and largely sea-ice covered polar regions due to sparse data) surface temperatures have exceeded previous

As above. Maybe change to ... For over a year since April 2023 () surface temperatures exceeded ...

44: coincides in time with atmospheric record warming

'Warming' implies that dT/dt was at a record. Do you mean: coincides in time with record atmospheric temperatures?

75: superimposed by
superimposed onto

91: Here we use observation-based monthly-mean SST estimates from various data products (NOAA OI SST V2.11, ERA529, HadISST30, ERSST31)

Here we use observation-based monthly-mean SST estimates from various data products (NOAA OI SST V2.11, HadISST30, ERSST31) and reanalysis (ERA529) ...

92: analyze the output from 270 simulations

analyze the output from 270 combined historical and future simulations

133: over a finite timeseries
over a short timeseries

134: so that the return period of such events might well be 1000 years as estimated from the models

I don't think this statement is useful. It might well also be 100 years. I would just delete

In Extended Data Table 1 there are some strange numbers (757,758,759) in large font near the end of the table

235: the also too high SST
...the overly high SST ...

240: over the following 8 years...
...within a few years ...

241: If this is also not the case
If this were not the case ...

244: Based on observations of past SST and using an AR(1) model

How about: Based on long synthetic timeseries with temporal characteristics that match available observations ...

246: in the context of human-induced climate change to date

Maybe: ...based on current warming rates ...

I would also break to 2 sentences. i.e. Such a jump would not have been possible without ...

249: by 0.25°C

...by at least 0.25oC

364: from the three reanalysis products ERA529, HadISST30, and ERSST31

Of these only ERA5 is a reanalysis (i.e. a model with data assimilation). The others are interpolated observational products

414: trend estimate to the estimation method

... trend to the estimation method

428: As it is not evident which reanalysis product

As above, only one of these is a reanalysis product

1 Response to reviewer 1

General Comments

Motivated by the unprecedented SST temperatures of 2023-24, this paper examines the likelihood of a warm global temperature record that exceeds the previous record by a large margin. The degree of exceedance from the previous record was unprecedented in the observational record. The study goes on to quantify the likelihood of such a high exceedance in various climate models. This question is of broad general interest, in particular whether such an event is consistent with our understanding of internal variability and whether climate models are able to capture such event.

Overall, the results will certainly be useful to inform the research community on the likelihood of the recorded sea surface temperatures. The analysis is sound and state-of-the-art. While the applied methods are maybe not groundbreaking, the study is very timely and thus of clear interest for a broad readership.

There are a few minor shortcomings in the discussion of the results, as well as aspects that the authors could have considered in more detail. I would recommend that they should consider these points in a revised version of the manuscript:

[...]

While these aspects could deserve more consideration in the article, this is a robust and helpful study for the research community, which could be considered for publication subject to minor revisions taking into account the above comments.

Response:

We thank the reviewer for their positive evaluation and the additional constructive and helpful comments, which have substantially improved our manuscript. We have taken each comment into account, provide responses to each point below, and adapted the manuscript accordingly.

Comment 1.1

The 2023 and 2024 temperature anomalies were not only extreme in terms of the SST anomalies, but also in terms of the land temperature anomalies. This paper only addresses the likelihood of the sea surface temperature anomalies, not that of the global mean temperature anomalies. Hence, it only addresses part of the picture. The authors could potentially pinpoint this in the article. It should also be mentioned that large anomalies in sea surface temperatures do not necessarily result in high land temperatures (e.g. Seneviratne et al. 2014, Nature Climate Change; Sillmann et al. 2014, ERL), hence it would not be substantiated to suggest that the land surface temperatures were the result of the sea surface temperatures. Possibly, the authors could also extend their study to assess the probability of the global mean temperature anomaly in the analysed climate models and whether their conclusions still hold.

Response:

As suggested by the reviewer, we have added the notion that land and sea temperatures are not necessarily related:

"This global record-shattering jump in SSTs ("record-shattering" is here defined as a record-breaking jump in annual globally averaged SSTs that exceeds previous records by at least 0.25°C, as observed in 2023/24) coincides in time with atmospheric record warming in late 2023²⁻⁵ and early 2024^{6,7}. Moreover, the record-shattering SSTs are believed to be responsible for the global atmospheric record warming^{4,5}, although temperature extremes over land and ocean are not necessarily related^{8,9}."

As the manuscript is focused on sea surface temperatures, we decided not to add the land temperature record to keep the manuscript short and focused.

Comment 1.2

b) The authors should be careful in some of their statements about the event not being the sign of unexpected, accelerated warming. Since the event is still on-going, if the sea surface temperature anomalies were to last much longer, this event would become at some point incompatible with the existing climate models. So, this conclusion is reasonable at the present point in time, but its long-term relevance will depend on the further evolution of the event in the coming 12-14 months. On the other hand, the article does mention what type of anomaly would no longer be consistent with climate models (an anomaly that would last longer than another 12 months) and this information will also be useful for the climate research community.

Response:

Following the reviewer's suggestion, we have added the following sentences to the revised manuscript to the results:

"As SST anomalies only do not return to pre-jump levels in these 'hot models'³⁰, it is likely that the global SST anomalies in the real world will return to where they were before the record-shattering jump in globally averaged SSTs. If, however, observed SSTs will not be coming back to pre-jump levels by September 2025, SSTs will likely warm slower in the years after 2025 so that SSTs will eventually return to the expected warming trajectory over the following 8 years (Extended Data Figure 5). Only in the unlikely case that SSTs do not return to pre-jump levels and warming will also not slow down so that SSTs will not return to the expected warming trajectory over the following 8 years, the ongoing extreme event would not be consistent with climate model simulations."

Comment 1.3

Abstract, lines 23-24: "and fueled weather extremes over land, for example causing extreme rainfall and droughts"

I would remove this statement since the authors do not seem to provide any evidence for this statement in the text and analyses. If anything, past events such as the so-called "global warming hiatus" have rather shown a decoupling between sea surface temperatures and hot extremes over land (Seneviratne et al. 2014, Sillmann et al. 2014).

Response:

The sentence has been removed as suggested by the reviewer.

Comment 1.4

Abstract, lines 36-37: "Thus, the global MHW in 2023 and 2024 is likely not a sign. . . *. Add "so far" before "not a sign". The authors highlight that this event a) is still on-going, and b) would be expected to return to pre-MHW levels between September of the 2nd year of the MHW and September of the following year; hence if the MHW continues beyond September of next year, the conclusion would be that this is actually an exceptional event that cannot be explained alone with the mean forcing so far.

Response:

Based on suggestions from reviewer 2, we have removed the notion of accelerated warming from the manuscript and adjusted the abstract substantially. In the new version, we have incorporated the here proposed suggestion by the reviewer as follows:

"These model simulations suggest that the record-shattering jump in surface ocean temperatures in 2023/24 was so far an extreme event after which surface ocean temperatures are expected to return to the expected long-term warming trend."

We have further made similar changes throughout the manuscript in accordance with the change in the abstract.

Comment 1.5

Abstract, lines 37-38: "but rather a very rare event"; add at the end of this sentence "that was made more probable because of human-induced global warming"

Response:

As the word count is limited in the abstract and given that we already have the information in the abstract that human-induced global warming was necessary for such an event to occur (see quoted sentence below), we decided to not add the words that the reviewer suggested.

"Without a global warming trend, such an event would have been practically impossible and can thus be entirely attributed to anthropogenic climate change."

Comment 1.6

Page 3, lines 50-51: "record-shattering is here defined as record-breaking global annual MHWs that exceed previous records by at least 0.25°C": Why this threshold of 0.25°C? Can this be justified with statistics?

Response:

The threshold of 0.25°C was chosen as it was the margin by which observed globally averaged SSTs in 2023/24 exceeded the previous record. We have chosen this threshold as this paper's topic is this observed event. We clarify this now in the revised manuscript:

"Overall, the annually averaged SSTs from April 2023 to March 2024 were 0.25°C larger (based on NOAA OI SST V2.1¹) than the previous record SSTs when averaged over the same months of a year. This global record-shattering jump in SSTs ("record-shattering" is here defined as a record-breaking jump in annual globally averaged SSTs that exceeds previous records by at least 0.25°C, as observed in 2023/24) coincides in time with atmospheric record warming in late 2023²⁻⁵ and early 2024^{6,7}. Moreover, the record-shattering SSTs are believed to be responsible for the global atmospheric record warming^{4,5}, although temperature extremes over land and ocean are not necessarily related^{8,9}."

Comment 1.7

Page 9, line 159: "that coincidence": the correct wording is likely "that coincide" here.

Response:

The sentence was changed as suggested by the reviewer.

Comment 1.8

Page 15, line 260: "was likely a 1-in-a-1000-year extreme event and not a sign of unexpected, accelerated warming": This wording could be misunderstood to mean that the MHW in 2023-2024 could have been induced by natural climate variability alone. However, the authors conclude clearly that this event can "entirely attributed to anthropogenic climate change". It is important to state that the event was a 1-in-a-1000-year extreme event in the context of the human-induced climate change so far, but does necessarily go beyond the expectations associated with the present climate conditions in relevant climate simulations. Maybe change the wording to "was likely a 1-in-a-1000-year extreme event in the context of human-induced climate change to date, which could not have happened without human influence, but is nonetheless at this stage not a sign of unexpected, accelerated warming."

Response:

The sentence does not exist in the revised manuscript. However, we have adjusted the new sentence in accordance with the reviewers' suggestion:

"Based on these idealized observation-based timeseries, the record-shattering global jump was a 1-in-760-year event (mean estimate) under the current long-term warming trend (1-in-295-year to 1-in-1800-year event; 95% confidence interval based on uncertainties from the observation-based trend, standard deviation, and autocorrelation, see Methods) (Fig. 2). Without human-made global warming, however, a record-shattering jump in globally averaged SSTs as observed last year would not have been possible as demonstrated by infinite return periods in idealised SST timeseries without an underlying trend (Extended Data Figure 2)."

Comment 1.9

Page 15, lines 261-264: "As climate models... they can be used...". While the authors have shown that the event so far is not incompatible with existing climate simulations, it was nonetheless extremely rare and is additionally still on-going. If it were to last much longer, there would be a point at which it would no longer be consistent with climate model simulations. The authors should be a bit more careful in formulating this conclusion.

Response:

Following the suggestion by the reviewers, we have changed the the manuscript and removed this part from the Conclusions. Instead it is now added to the last paragraph and before the Conclusions and reads as follows after accounting for the reviewers' suggestions:

"As SST anomalies only do not return to pre-jump levels in these 'hot models'³⁰, it is likely that the global SST anomalies in the real world will return to where they were before the record-shattering jump in globally averaged SSTs. If, however, observed SSTs will not be coming back to pre-jump levels by September 2025, SSTs will likely warm slower in the years after 2025 so that SSTs will eventually return to the expected warming trajectory over the following 8 years (Extended Data Figure 5). Only in the unlikely case that SSTs do not return to pre-jump levels and warming will also not slow down so that SSTs will not return to the expected warming trajectory over the following 8 years, the ongoing extreme event would not be consistent with climate model simulations."

2 Response to reviewer 2

General Comments

Motivated by the unprecedented SST temperatures of 2023-24, this paper examines the likelihood of a warm global temperature record that exceeds the previous record by a large margin. The degree of exceedance from the previous record was unprecedented in the observational record. The study goes on to quantify the likelihood of such a high exceedance in various climate models. This question is of broad general interest, in particular whether such an event is consistent with our understanding of internal variability and whether climate models are able to capture such event. I do have a number of general concerns however:

Response:

We thank the reviewer for their positive evaluation and the additional constructive and helpful comments, which have substantially improved the manuscript.

Most importantly, we have changed the wording throughout the manuscript by not calling the here analysed events marine heatwaves anymore and we now estimate the return period based on observations and a synthetic timeseries using an autoregression model of first order (AR1).

Furthermore, we have taken each comment into account, provide responses to each point below, and adapted the manuscript accordingly.

Comment 2.1

First is the writing in the document. I found it very hard to follow what the authors were doing in many parts of the manuscript, in particular I feel that the description of the analysis could be significantly improved. The reader loses track of the fact that you are focussing on record events with unprecedented increment in SST, not just really big events. I also find it unhelpful to call these events MHWs – you are examining a global and annual average, there's no reason to expect that all regions experience MHW conditions for all or part of the year. To help with the narrative of the manuscript, it might be helpful to set out your questions up front e.g.: how likely is this kind of event? Would it be possible without anthropogenic warming? Are models able to simulate similar events? What can models tell us about the causes of such events? Could this event be the start of a regime shift to permanently warmer temperatures? Are such events likely to reoccur in the future? This way you have a clear motivation for the particular analysis that you carry out. The only question you articulate at the moment “whether this still ongoing event indicates an unexpected acceleration of global warming or simply a very rare extreme event” isn't actually answered by your study.

Response:

As suggested by the reviewer, we have removed the term marine heatwave from almost all parts of the manuscript. Instead we now talk about record-shattering jumps in globally averaged SSTs that exceed previous records by at least 0.25°C (the margin by which the observed SST record was broken in 2023/24). We have also set out the questions as suggested by the reviewer at the end of the Introduction in the revised manuscript:

"Here we use observation-based monthly SST estimates from various data products (NOAA OI SST V2.1¹, ERA5²⁷, HadISST²⁸, ERSST²⁹) and analyse the output from 270 climate model simulations to quantify the likelihood of the global SST jump in 2023/24 and assess whether it could have occurred without anthropogenic warming. The model simulations also provide insights into the spatial pattern of such events and how the currently ongoing event may evolve in the future."

Comment 2.2

It seems that the primary goal of this study is to establish the return period of an event like the one that occurred in 2023/24. Given the significant biases in the ability of models to realistically simulate aspects of internal variability (something that is discussed in the literature, but hasn't really been discussed in this manuscript), I'm not sure that the proposed method is the best. For example you could simply construct a long synthetic timeseries based on observed temperature and establish the return period. You could then usefully use the large ensembles to estimate uncertainty bounds in the estimate of observed variability.

Response:

As suggested by the reviewer, we now establish the return period of an event like the one that

occurred in 2023/24 using a long synthetic timeseries using an autoregression model of first order (AR1).

To account for the reviewers suggestion and in light of the new results from the AR1 model, we now discuss differences in the climate variability, warming trends, and autocorrelation between the models and the observation-based estimates in an entirely new paragraph in the methods. In addition, we discuss the effect on the estimated return period and differences between the model estimates and observation-based estimates for the return period:

"The climate model simulations analysed here have temperature trends, temperature variabilities and autocorrelations that spread around the observation-based estimates of these three variables (Extended Data Figure 2). The temperature trend from 2004 to 2023 in the models is 20% smaller than the observation-based trend estimated from NOAA OI SST V2.1¹ (0.25°C per decade) in CMIP6 model simulations, 21% smaller than the observation-based trend in CESM2 simulations, and 47% smaller in GFDL-ESM2M simulations. The standard deviation of the temperature variability (temperature from 1940 to 2023 after being detrended with a 3rd order polynomial) is 16% higher in CMIP6 model simulations than the average of the observation-based estimates based on ERA5²⁷ (0.085°C), HadISST²⁸ (0.082°C), and ERSST²⁹ (0.092°C), and 15% and 20% higher than the observation-based estimate in CESM2 and GFDL-ESM2M simulations, in line with the overestimation of decadal trends of major climate modes in CMIP6 models³⁹. In addition, the autocorrelation in CMIP6 models is 55% larger than the average of the observation-based estimates based on ERA5²⁷ (0.20), HadISST²⁸ (0.31), and ERSST²⁹ (0.41), 23% larger in CESM2 simulations, and 19% larger in GFDL-ESM2M simulations. A part of the difference between the models' and the observed parameter estimates may be due to the uncertainty in the observed estimates (standard deviations for the parameter estimates of SST trend, SST standard deviation, and year-to-year SST autocorrelation of 0.03 °C dec-1, 0.004 °C, and 0.06, respectively). In addition, the counting of the record-shattering jumps in globally averaged SSTs led to a return period of 1000 years, slightly higher than the mean estimate of the idealized timeseries (850 years with a 95% confidence interval of 295 to 1800 years) but still well within the confidence interval. While the return period in climate models (1000 years with a 95% confidence interval from 550 to 2000 years) is well inside the confidence interval of the observation-based estimates, the slightly higher simulated return period might well be the result of the slight tendency towards relatively high autocorrelations and relatively small trends, compensated by a relatively high standard deviation."

Unfortunately, we are not sure to which literature the reviewer is referring to with respect to internal variability in CMIP6 models. As opposed to the reviewers suggestion that internal variability is underestimated in CMIP6 models, Extended Data Figure 2 in our manuscripts suggests that the standard deviation of SSTs is often higher in CMIP6 models than in observations. Moreover, Terhaar et al. (2024) also showed that the variability and decadal trends in climate modes such as the Southern Annular Mode, the Atlantic Multi-decadal Oscillation, or the El Niño 3.4 index are larger in CMIP6 models than in observations. We are, however, more than happy to incorporate additional literature to the Discussion if the reviewer suggested some references.

Comment 2.3

Moreover, using white noise to simulate the variability in annual temperatures is possibly over simplistic. The analysis around extended figure 2, for example indicates memory in temperature from one year to the next. The distribution of annual temperatures may also not be gaussian (was this tested?). This requires more careful consideration (e.g. you might consider an AR model). At least some sensitivity analysis should be made to your choices.

Response:

We thank the reviewer for this great suggestion. We have now replaced the simple model based on trends and Gaussian white noise by an AR1 model. The auto-correlation has a large effect on the return period with higher auto-correlation leading to a longer return period. These new results are now carefully implemented in the manuscript. Figure 4 in the old manuscript was replaced by Figure 2 showing the distribution of the return period estimates. In addition, an update of Figure 4 (now Extended Data Figure 2) that includes a range of different auto-correlations has now been added. The new results were added throughout the manuscript and the observation-based return period is now estimated based on the AR1 framework and highlighted by a separate figure (Now Fig. 2).

As in the previous approach, we also assume a Gaussian distribution for the data in the AR1 model. We now add Kolmogorov Smirnov tests to the revised manuscript to test if the distribution of temperature anomalies is Gaussian. We find p-values of 0.78, 0.91 and 0.49 for ERA5, HadISST, and ERSST, respectively. As the hypothesis that the distribution is non-Gaussian cannot be confirmed, we conclude that the Gaussian distribution is a reasonable assumption.

Comment 2.4

It would have been good to see some speculation, guided by the models, as to why this very rare event occurred. You showed for example the importance of an El Nino (but this isn't surprising as record warm events are typically associated with El Nino). But what causes such a large deviation from previous records. The models should be able to help answer this question.

Response:

The models should indeed be able to answer these questions. As mentioned by the reviewer, we have only slightly touched the question why this event has occurred. In this manuscript, however, we have focused on the questions mentioned by the reviewer above: return period, dependence on anthropogenic warming, common patterns, the outstanding jump in observed SSTs in the North Atlantic, the duration of the events, and the question if models can be used to analyse such events. As the length of the manuscript is limited and a driver analyses would exceed the scope of the study, we hope that our study will show the wider community how useful the models are indeed and hence motivate colleagues to use these models to find answers to this highly important question.

Comment 2.5

33. In climate models, sea surface temperatures . . . Hard to understand what you mean here without more information. I suspect this will become clear after reading the manuscript, but its not from the abstract alone

Response:

The mentioned sentence does not exist anymore in the revised manuscript.

Comment 2.6

44. resulting in a global marine heatwave . . . Just because the global average reached record temperatures does not imply MHW conditions globally (i.e. at all locations). I don't feel that this terminology is accurate

Response:

As suggested by the reviewer, we have removed the terminology throughout the manuscript when referring to the record-breaking jump in globally-averaged SSTs.

Comment 2.7

51: has been accompanied by atmospheric record warming > was accompanied by record atmospheric warming

Response:

This part of the sentence has been removed following a large change of this sentence as suggested by reviewer 1.

Comment 2.8

78: The record-shattering global MHW in 2023 and 2024 has raised the question if this global MHW is a very rare extreme event or if climate change and global warming have been accelerating faster than expected based on fully coupled climate model simulations. We already know that this is a rare event (its “record shattering”). And a rare event can be caused by anthropogenic warming, so the either/or doesn’t make sense. “climate change and global warming” what’s the difference? I don’t see why you need “based on fully coupled climate model simulations” – you are interested in these questions irrespective of whether it’s consistent with climate models. This framing needs some work. How about: This raises questions around whether this event can be explained by natural variability alone, whether and how much it is affected by anthropogenic warming. And whether the extremeness of this event is unexpected even given the background warming that has occurred, based on comparison with climate models. If this is indeed the case it would suggest important shortcomings in the ability of our models to ...

Response:

Following the reviewers’ suggestion, we have substantially changed the framing of the manuscript to:

"While record-breaking jumps in global SSTs can occur when the long-term warming trend is superimposed by an exceptionally warm year due to climate variability²⁴, the magnitude by which the record-shattering jump in globally-averaged SSTs in 2023 and 2024 has broken previous records came as a surprise for the public and the scientific community. This event has raised questions about the likelihood of such a jump and whether this jump could have been expected based on fully coupled climate model simulations²⁵. If state-of-the-art fully coupled climate models were not able to reproduce such events, the jump in SSTs in 2023/24 would consequently question the ability of these models to assess future risks associated with anthropogenic climate change²⁶. However, if fully coupled climate models were able to simulate record-shattering jumps in globally averaged SSTs as the one observed over the last year, they would deliver analogues to study the evolution of the ongoing event and to project how SSTs continue to evolve after such events. In addition, the climate models could be used to identify the drivers of such record-shattering jumps in SSTs, and their consequences for other parts of the climate system and the marine ecosystem."

In addition, we now explicitly address the questions around whether this event can be explained by natural variability alone and whether and how much it is affected by anthropogenic warming.

Comment 2.9

79: accelerating faster than expected You are describing a single event, I don't see that you can look at acceleration of warming rates based on a single year. I think this should be framed in terms of how exceptional the event was/is such an extreme event consistent with our models? Are the models missing something? NB even if this scale of event is unprecedented in models, that doesn't imply global warming is accelerating. Its more likely that the models are underestimating the degree of internal variability – something that has been illustrated in many prior studies If the goal is to see if the observations are accelerating faster than climate models, why wouldn't you just compare the acceleration of the warming in model and observations. Why go to the trouble of looking at differences in return periods of extreme events?

Response:

As suggested by the reviewer, we have largely removed the notation of accelerated warming and now focus on the ability of models to simulate such events and if they simulate these events for the correct reasons (magnitude of trends, variability, and autocorrelation). The changes were made throughout the entire manuscript.

Unfortunately, we still do not know which literature the reviewer is referring to with respect to climate variability in CMIP6 models and refer to our response to Comment 2.2.

Comment 2.10

82: be in uncharted territory ... The event was unprecedented in the last 2000, so its already uncharted territory.

Response:

Following this and previous suggestions by the reviewer, this sentence has been removed from the manuscript.

Comment 2.11

83: you don't need the 'so far'. Projections are the output of climate models

Response:

"so far" was removed from the manuscript as suggested by the reviewer.

Comment 2.12

114 global (60°S-60°N) This is not global

Response:

The 60°S-60°N range is indeed not global. However, for sea surface temperatures, it is not possible to consistently measure sea surface temperatures in polar regions due to sea ice and dense cloud cover. Due to the data availability in high-latitude oceans and given that 60°S-60°N covers 89% of the global ocean surface, global SSTs are often defined as SSTs between 60°S and 60°N (see for example https://climateranalyzer.org/clim/sst_daily/?dm_id=world2). We have changed the sentence to:

"Since April 2023, global ocean (60°S-60°N; excluding cloudy and largely sea-ice covered polar regions due to sparse data) surface temperatures..."

Comment 2.13

Figure 2. You haven't described what the figure is showing. My guess is that it's the temperature difference relative to the next highest record temperature, but this is not clear from any of the description.

Response:

As suggested by the reviewer, the figure legend was changed to:

"Figure 3: Record-breaking annual sea surface temperature events from 1965 to 2040, the margin by which they exceed previous records, and their return periods. a) Record-breaking annual (April-March) global (60°S-60°N) sea surface temperature (SST) events and the margin (Δ SST) by which they exceeded previous records based on 190 climate model simulations from CMIP6 (blue dots), 30 members of the GFDL-ESM2M-LE (orange dots), 50 members from the CESM2-LE (green dots), and SST observation-based estimates from NOAA OI SST V2.1¹ (black crosses)..."

In addition, we changed the title in label a) to "Margin of record-breaking SST event (Δ SST)" to better explain what the figure shows.

Comment 2.14

125: Record-breaking heatwaves that are at least as large in magnitude as the record-shattering MHW over the last year were simulated 11 times between 2000 and 2040 This description is very confusing. If I understand correctly, you are not examining the magnitude of the event you are examining the difference in magnitude from the previous record. Maybe, you could call it something like a 'record increment' so its clear what you mean

Response:

As suggested by the reviewer, we do not call such events MHWs anymore. Instead, we now call them jumps in globally averaged SSTs that exceed previous records.

Comment 2.15

127: surrounding the year This would be 2003-2043

Response:

We have changed the sentence to:

"These four decades encompass the time when the record-shattering SST jump occurred in the real world (Fig. 3a, Extended Data Figure 1)."

Comment 2.16

131: making last year's record-shattering MHW a 1-in-1000-year event This is if we believe the climate models. To do this we would want to see that the models have a trend and variability that is consistent with the observations.

Response:

Following previous suggestions by the reviewer we now derive the return periods based on an AR1 model and observation-based estimates of the variability, autocorrelation, and trend. In addition, we have changed the sentence mentioned by the reviewer as follows:

"As the 270 climate model simulations here comprise a total of 11,070 years between 2000 and 2040, the likelihood for a single year to experience a record-shattering global jump in SSTs as observed in 2023/24 in the climate models is 0.1%, making last year's record-shattering SST event a 1-in-1000-year event in climate models (1-in-550-year to 1-in-2000-year event; 95% confidence

interval; see Methods) (Fig. 3b)."

Moreover, we now expanded Figure 4 (now Extended Data Figure 2), and added an entire paragraph to compare the trends, variability, and autocorrelation from the models to the respective observation-based estimates to the Methods.

Comment 2.17

134: 74 times Makes more sense to talk about a percentage

Response:

Following the suggestion by the reviewer, we have, however, extended the sentence as follows:

"Jumps in globally averaged SST that break the old record by a smaller increment occur increasingly often, for example 74 times between 2000 and 2040 for jumps in globally averaged SSTs that break the old record by more than 0.2°C (Fig 3b), approximately 6-7 times more often than jumps that break the old record by 0.25°C."

Comment 2.18

136-141: The climate models that simulate most of these record-shattering global MHWs (>0.25°C) are mostly also the ones with the most ensemble members This isn't really an interesting result. If record breaking events are evenly distributed, then we would expect models with more ensembles to have more records. More interesting is whether the proportion of records per ensemble is particularly unusual in certain models

Response:

As suggested by the reviewer, we have moved part of this paragraph to the Methods. The new paragraph now reads:

"The climate models that simulate most of these record-shattering global jumps in SSTs (>0.25°C) are not only so called 'hot models'³⁰ with high transient climate responses, like IPSL-CM6A-LR, CESM2, and CanESM5. Models with a small transient climate response and warming trend, such as GFDL-ESM2M and MIROC6, also produce these record-shattering jumps since variability and autocorrelation also affect the return period, and not just the long-term trend (Extended Data Figure 2). In general, such extreme events are simulated by models that provide large ensembles and no model ensemble stands out with an unusually small or high number of extreme events (see Methods)."

Although it may not sound like an interesting result, we feel that it is still important to show

that there is no particular unusual proportion for any model and that such events in models are thus not a result of the 'hot model' problem²⁸. Given that the observation-based estimate of the return of record-shattering jumps in SSTs has a confidence interval between a 1-in-a-290 year event and a 1-in-a-1800 year, it is not unusual that no such events are sampled in model ensembles with 10 or less ensemble members, which cover a maximum of 410 years. The smallest ensemble in which an event was simulated was IPSL-CM6A-LR, which samples 246 years. Finding such an event in one small ensemble is also not unusual given that there are many such small ensembles without a record-shattering SST jump. Lastly, the most events were sampled in the MIROC6 ensembles with 4 events in 1763 years (43 ensemble members). This corresponds to a return period of around 440 years, within the uncertainties of the observation-based estimates.

Comment 2.19

Figure 3. The caption could be written more clearly. You are simply showing the April-march average temperature anomaly, relative to the preceding 30 years . . .

Response:

Following the suggestion by the reviewer, we have changed the caption to:

"Maps of sea surface temperature anomalies for record-shattering events in observations 2023/2024 and one specific member of the three climate model groups that contain such events before 2023/2024".

Comment 2.20

146-148: what evidence do you have for these statements?

Response:

The previous sentence refers to Fig. 3a where observation-based estimates of SSTs are shown based on NOAA OI SST V2.1. This same data was analyzed to provide evidence for this statement. We have now added a reference to this dataset directly into the paragraph and not only to the figure.

Comment 2.21

150: all record-shattering MHWs coincide with a positive El-Niño phase Do you mean all the 11 events in the climate model simulations are associated with El Nino?

Response:

Yes, all 11 events are associated with an El-Niño. The manuscript has been adjusted accordingly:

"In climate models, events as large as that in 2023/24 all coincide with an El-Niño event, i.e., a positive El Niño-Southern Oscillation phase of at least 1.5°C in the El-Niño 3.4 index (Fig. 4b-d). As the three most recently observed record-breaking jumps that broke the respective previous record in globally averaged SSTs by unprecedented margins (2023/24, 2015/16, and 1997/98) also occur during positive El-Niño phase (1.3°C, 1.9°C, and 1.7°C respectively), a strong El-Niño appears to be a necessary, but not sufficient condition for such an event."

Comment 2.22

151: 2015/16, and 1997/98 So these years had the largest increments from the previous records?

Response:

Yes, they have the largest increments. To avoid any further confusion, the sentence was changed as shown in the response to comment 2.21 just above this comment.

Comment 2.23

160: did not break the old record by the same margin . . . This is the kind of terminology that you should be using throughout. Its far less ambiguous than the terminology used elsewhere.

Response:

As suggested by the reviewer, we have made this change throughout the entire manuscript and removed the notion of MHWs.

Comment 2.24

162-175: As global and North Atlantic record-shattering MHWs are extremely unlikely and global record-shattering MHW patterns can vary in magnitude, an exceptionally large global MHW that occurs at the same time as an even less likely record-shattering MHW in the North Atlantic is even more unlikely. I don't really see what point you are trying to make. High SSTA in the north Atlantic would make it more likely that the global average SSTA will be higher. But the NA only makes up a small part of the global ocean so this result is completely expected. Im not sure what value it adds to this study. Also if you are talking about the NA why not also the north Pacific or eastern tropical Pacific too?

Response:

We are discussing the North Atlantic as this region was especially warm in the observed record-shattering jump in SSTs in 2023/24 and SSTs in the North Atlantic exceeded the previous record in that region by 0.42°C , a margin that is 0.17°C larger than the margin by which global SSTs were broken. Furthermore, the other two regions with relatively large SST anomalies in 2023/24, the North Pacific and Eastern Tropical Pacific, did not break SST records. We have adjusted the manuscript to better explain why we pay special attention to the SSTs in the North Atlantic.

Comment 2.25

191: 13 to 18 months after the beginning of the MHW... This suggests that the ocean has a memory that exceeds 1 year timescales. This would suggest that your assumption of independent years for your synthetic timeseries does not really hold.

Response:

As mentioned earlier, we now construct the synthetic timeseries with an AR1 model to account for that memory.

Comment 2.26

194: Only in 3 of the 11 record-shattering global MHWs, SST anomalies do not return to pre-MHW levels In all the models there are some green lines that dip below the orange line. So this statement isn't completely accurate. Also you are only looking at the next 10 years, so your statement only applies to the following decade.

Response:

We believe that this is a misunderstanding. If green lines drop below the orange line, this means that SSTs are not anymore record-breaking. To come back to pre-jump levels (or pre-MHW levels in the previous version of the manuscript), the green line would have to drop below a grey line. This is not the case in 3 out of 11 events. To avoid further confusion, we now add that the grey lines indicate the pre-jump condition.

As the scenarios here are SSP5-8.5, RCP8.5, and SSP3-7.0, which all show strong warming trends, it is impossible that they drop below after more than a decade (as can also be seen by the new Extended Data Figure 5 for three examples). We have also tested this. We believe that the statement is sufficient and that it is not necessary to show a plot with the respective timeseries.

Comment 2.27

198: as indicated by the fact that the distances between the SSTs over the years . . . If there were say a 10% increase in the background warming rate, do you think you would be able to spot it just by looking at the distance between lines? Im skeptical.

Response:

Following the reviewers comment, we now compare these simulations to their respective ensemble mean, added a corresponding new figure in the Extended Data (Extended Data Figure 5 in the revised manuscript), and changed the sentence accordingly:

"In only in 3 of the 11 events, the SST anomalies do not return to pre-jump levels (Extended Data Figure 4), so that the global SSTs have indeed permanently risen to a higher level in these simulations. However, even in these three cases, the warming trend after the jump in SSTs has not accelerated as these models' SSTs warm less fast in the years following the jump and return to the respective ensemble mean within 8 years after the event (Extended Data Figure 5). "

Comment 2.28

201: The 3 cases where the gap is not filled again are simulated by CanESM5 and CESM2 . . . Which figure panels are you referring to? The clearest case for hot temperatures after the extreme year looks to be the IPSL model (panel d). CanESM2 has 2 ensembles one where it stays relatively hot, the other where it doesn't. Also looking at fig 4, it looks like the CESM2 ensemble has a systematically weak trend My guess would be that its more about the strength of internal variability in the model rather than the warming trend.

Response:

CESM2 is indeed not one of these models. Instead IPSL-CM6A-LR should have been mentioned

here. This mistake was corrected in the revised manuscript.

With respect to the two CanESM5 simulations, SSTs in one simulation never get close again to pre-jump levels. In the other simulation (Extended Data Figure 4b), the SSTs come back very close to pre-jump levels in November and December but never completely back but the gap is more visible in annual averages in the now added new Extended Data Figure 5.

We also analyzed the standard deviation and autocorrelation of the temperature variability, as well as the temperature trend from 2004 to 2023. All standard deviations in these 3 simulations where SSTs do not return to pre-jump levels are 3-15% larger than the average of the observation-based estimates, the trends are between -12 and +30% of the observation-based estimate, and the autocorrelations lie between -3 and +117% of the observation-based estimate. It is thus not straight-forward to identify one main reason, especially given that the observation-based estimates have large uncertainties attached to them (the autocorrelation estimate in ERSST, for example, is 100% larger than the estimate based on ERA5). As the 20-year trend is also highly uncertain, we believe that the high TCRE and ECS is the likeliest reason for the SSTs to not come back and keep the text as it is.

Comment 2.29

Figure 4. Its important to remember that there will be a large spread in the observational estimate, because of the length of the timeseries used to calculate the trend or the standard deviation. You can see this from the large ensembles. A single model (like the CESM2) will have the same forced trend and variability, but the estimated trend and variability differ because you are only using a few years to calculate these metrics. For example from the CESM ensemble Id guess that there's an error of about +/-0.1oC/decade in the trend. (Indeed this might be a helpful to put error bounds on the observational return period)

Response:

Following this comment and previous comments by the reviewer, we now estimate the return period in the revised manuscript from observation-based estimates and clearly state the uncertainties in the return period estimate that arise from uncertainties in the observation-based estimates of trend, variance, and autocorrelation (Figure 2 and Extended Data Figure 7).

Comment 2.30

229: best represent Best compared to what? All other models? This wouldn't be supported by your analysis as there are blue circles from individual models that are closer to the observational estimate. If you are comparing to the MMM I suspect that the CMIP6 MMM trend is closer to the observational estimate than the ESM2M model. Indeed the spread in the ESM3M model does not even overlap with the observations, which suggests to me that this wouldn't be a good representation of the real world. The reason that the return period isn't too bad is because of compensating errors – the trend is too weak, and the variability is too high.

Response:

Based on the reviewers comment, we have removed the sentence from the revised manuscript.

Comment 2.31

260: was very likely a 1-in-a-1000-year extreme event This model based estimate, is based on models that are clearly biased (based on your fig 4). Would a more reliable estimate be obtained using just the observational data to construct a long synthetic timeseries? The large ensembles could then be used to provide an estimate of uncertainty on the estimated variability. Assuming that variability is stationary at different levels of warming, the return rate would then depend on the rate of global warming.

Response:

We now estimate the return period with an AR1 processes and observation-based estimates of the variability, autocorrelation, and trend as suggested by the reviewer. The estimated return period is similar to that of the models.

Comment 2.32

231: The slight overestimation of the return period I don't really see an overestimation. If you draw a line of constant return period through the observations, there seems to be almost equal numbers of models above and below.

Response:

The overestimation in the last version of the manuscript was concluded by comparing synthetic timeseries based on observation-based estimates of trend and variability and synthetic timeseries based on simulated trend and variability. As the synthetic timeseries have now been changed, the

sentence does not exist anymore in the revised manuscript.

Comment 2.33

235: only the models with high transient climate responses, such as IPSL-CM6A-LR, CESM2, and CanESM5 As above, for the two large ensembles, your analysis of fig. 4 points to an overly weak trend not an overly strong trend, over the decades considered)

Response:

The observation-based estimate of the trend, calculated as a linear trend from 2004 to 2023, has large uncertainties, mainly due to outliers at the beginning and the end of the timeseries. As such, the observation-based estimate of the trend might be on the upper end of the uncertainty range. However, transient climate responses are not subject to those uncertainties as they are calculated as the average warming over 20 years after CO₂ has been doubled in the atmosphere. As a 20 year mean is much less sensitive to internal variability than a linear trend over 20 years, we discuss the transient climate response and not trends here.

In the estimate of the return period, we now include these large uncertainties of observation-based estimates of the variability, the recent trend, and the autocorrelation (see Methods, Figure 2, and Extended Data Figure 7).

Comment 2.34

343: Im not sure that ERA5 SST is the best choice as it can contains inhomogeneities in time. Products like HasISST or ERSST are designed for looking at temperatures over long timescales

Response:

We have used ERA5 as it is one of the most often used products. However, following the reviewers suggestion, we now added the two proposed products. We now use these different products together to estimate uncertainties of the observation-based estimates of the variability and autocorrelation (see revised Methods). The trend is still estimated from NOAA OI SST V2.1 as it is the best SST estimate that exists for the last 4 decades.

Comment 2.35

352: You should indicate the number of different models

Response:

As suggested, we added the number of different models.

Comment 2.36

367-368: Im not clear why you would even talk about re-gridding as you are dealing with global or regional averages

Response:

We feel it important to mention that we did not regrid, because many multi-model studies regrid the data first. This information is important to be able to reproduce the study.

Comment 2.37

379 onwards: I found it difficult to follow what you are doing here. The description needs significant improvement. I think a large part of the problem is that you don't say what you are calculating the return period of. You are looking for events where the globally averaged 2023 Arpril-2024 March is the hottest on record and the record exceeds the previous record by at least 0.25oC.

Response:

The section was entirely changed in the manuscript because we now use an AR1 model. We made an effort to write the new section as understandable as possible.

Comment 2.38

380-390 I don't find this very clear. I think that you are constructing synthetic timeseries with different standard deviation and trend to look at how often a new record temperature exceeds the previous record by at least 0.25C

Response:

The paragraph does not exist anymore in the revised manuscript.

Comment 2.39

387: Your extended figure 5 looks very bimodal, not gaussian. There are quantitative metrics to determine how close distributions are to being gaussian, rather than just looking at this by eye.

Response:

We are not aware of a good reason to assume a bimodal distribution for global annual SSTs - as such, and given the consistency of the data with a Gaussian distribution that was tested with a Kolmogorov Smirnov test in the revised manuscript, we believe it is more sensible to assume a Gaussian distribution for the synthetic time series.

Comment 2.40

389: Have you looked at the temporal autocorrelation to check how independent consecutive years are. I suspect that your results are going to be sensitive to the assumptions you make about the theoretical timeseries. I suspect some form of autoregressive model would be more appropriate.

Response:

As suggested by the reviewer, we made major adjustments in the revised manuscript by now considering the autocorrelation using an AR1 model.

Comment 2.41

391: The theoretical return period for observations and climate models Not sure what you mean by this. Do you mean that you calculate the return period from a long synthetic timeseries with the same 2004-2023 trend and 1940-2023 standard deviation as the observations? Id be sceptical of using a 20 year trend as a proxy for anthropogenic warming, if that is what you are aiming to do.

Response:

Yes, we meant the return period based on a synthetic timeseries. As it caused confusion, we have clarified this wording.

In addition, we believe that the 20 year trend is indeed not a good proxy for anthropogenic warming as suggested by the reviewer. However, the likelihood of the record-shattering jump in SSTs in 2023/24 does not depend on anthropogenic warming but on the actual temperature evolution, which is also influenced by volcanoes and internal variability. It is thus this trend over these last 20 years that determined the return period of such an event in 2023/24.

This trend in observed temperatures over the last 20 years comes with large uncertainties that translate into large uncertainties in the return period. This is now part of the estimate of the return period uncertainty in the revised manuscript. Ideally, a longer time period would be used to estimate the trend to reduce this uncertainty. Unfortunately, the warming trend has largely accelerated after 2000 and been relatively linear since then so that there are not more years that could be used. Similarly, the autocorrelation and variance estimates also comes with large uncertainties, despite being estimated from three products and over a much longer period of 84 years.

Comment 2.42

396: by subtracting a 3rd order polynomial Why? Are your results sensitive your assumed shape of anthropogenic warming. E.g. do your results change if you use a linear fit or 2nd order polynomial

Response:

As suggested by the reviewer, we tested the sensitivity of the estimated variance and autocorrelation to the choice of using a 3rd order polynomial to isolate the variability by removing the underlying trend from the SST timeseries. However, given the non-linearity of the timeseries of SSTs from 1940 to 2023, we believe that a linear fit is not an adequate choice. Instead, we use an Enting spline here (Enting et al., 1994) with a cut-off period of 50 years, long enough to not remove decadal variability as well, in addition to the 2nd order polynomial. Here we show the values from the 2nd order polynomial and from the Enting spline fit are similar to those from the 3rd order polynomial fit (the first value is for the 2nd order polynomial, the second value for the Enting spline fit and the values in parenthesis are the respective values when using a 3rd order polynomial):

Variance:

ERA5: 0.0072 / 0.0072 (0.0072) °C²

ERSST: 0.0091 / 0.0079 (0.0084) °C²

HadISST: 0.0069 / 0.0065 (0.0067) °C²

Mean: 0.0077 / 0.0072 (0.0074) °C²

Autocorrelation:

ERA5: 0.20 / 0.19 (0.20)
ERSST: 0.47 / 0.38 (0.41)
HadISST: 0.34 / 0.29 (0.31)
Mean: 0.35 / 0.29 (0.34)

Based on this analysis, we changed the manuscript as follows:

"The variance and autocorrelation for each of the three observation-based timeseries of globally averaged ocean SSTs were both calculated after detrending by subtracting a 3rd order polynomial that was fitted to the respective SST timeseries. Using a 2nd order polynomial or an Enting spline for detrending results in qualitatively and quantitatively similar results of the mean variance and autocorrelation within $\pm 10\%$ when compared to results when using a 3rd order polynomial. As such, the mean estimates based on 2nd order and Enting splines diverge only little from the mean estimates based on the 3rd order polynomial compared to the sampling uncertainty in the mean estimates (Extended Data Figure 7) that is taken into account to estimate uncertainty in the return period. Furthermore, the 3rd order polynomial-based estimates are located between those from the 2nd order polynomial and Enting splines, overall suggesting that the chosen detrending methodology neither results in additional uncertainty nor a systematic bias in the variance and autocorrelation estimates. As the 3rd order polynomial-based estimates lie in between the estimates from the 2nd order polynomial and the Enting spline, we believe that the 3rd order polynomial-based estimates provide the best estimates of the mean values."

Enting, I. G., Wigley, T. M. L., and Heimann, M.: Future Emissions and Concentrations of Carbon Dioxide: Key Ocean/Atmosphere/Land Analyses, 1994.

Comment 2.43

Table 1 & 2 could be summarised, just give the number of ensembles per model. (If the idea was to make the results reproducible you would also need to include simulation version numbers/download dates etc.)

Response:

As suggested by the reviewer, we have summarized table 1 and incorporated the information from table 2 into the Methods.

Comment 2.44

398: 321 years for the CMIP6 ensemble, 596 years for the GFDL-ESM2M-LE, and 377 years for the CESM2-LE If the observed return period is 440. Then 321 is not 30% of 440 and 596 is more than 100% of 440, so Im not sure what these numbers mean.

Response:

This sentence does not exist anymore in the revised manuscript.

Comment 2.45

407: Again its hard to understand what you are doing.

Response:

We have added a reference from a standard text book and now also discuss in the revised manuscript the robustness of our confidence interval estimate:

"For the climate model return period estimate, a confidence interval for the return period estimate of record-breaking global jumps in SST that exceed the previous record by at least 0.25°C was constructed by identifying the counted number of such events in the model simulations (here 11) with the outcome of a binomial experiment⁴⁷. The binomial parameter p represents the probability of any year in any model simulation to show such an event. We then calculate the 95% Pearson-Clopper confidence interval (p_{lower} , p_{upper}) for the binomial parameter p ^{47,48}. As the return period is given by $1 / p$, the confidence interval of the return period is $(1 / p_{\text{upper}}, 1 / p_{\text{lower}})$.

The Pearson-Clopper confidence interval was chosen over that assuming a normal distribution for p , since p is close to zero where normality can not be assumed⁴⁷. Nonetheless, a relatively similar confidence interval (633-2459 years) would be estimated when assuming a normal distribution. This similar result suggests that the confidence interval is relatively insensitive to the chosen approach."

Comment 2.46

405: In what way is 440 qualitatively similar to 321 or 596?

Response:

This paragraph does not exist anymore in the revised manuscript.

1 Response to reviewer 2

General Comments

This is a re-review of Likelihood of the record shattering jump in surface ocean temperatures in 2023 and 2024. First off I'd like to thank the authors for diligently responding to all my comments. I think the paper is much improved. I do still have some comments/concerns that I feel need addressing. I've also made a number of text modifications to help with readability.

Response:

We thank the reviewer for their positive evaluation and the additional constructive and helpful comments and modifications that substantially improved the manuscript's readability and the robustness of the applied methods. We have taken each comment into account, provide responses to each point below, and adapted the manuscript accordingly.

Comment 2.1

20: exceeding previous records
I think you mean: 'exceeding the previous record'
Also it would be worth stating when that record was set.

Response:

Following the suggestion by the reviewer, we have changed the sentence to:

"Global ocean surface temperatures have been at record levels for more than a year since April 2023, exceeding the previous record from 2015/16 by 0.25°C on average between April 2023 and March 2024."

Comment 2.2

25: by 0.25°C presumably "by at least 0.25oC"

Response:

Changed as suggested.

Comment 2.3

27: been practically impossible and can thus be entirely
if its 'practically impossible' then its 'almost entirely' not 'entirely'!
Also I don't agree with this attribution statement. If its entirely related to AGW that means
that it would have happed even without the extremely rare internally generated jump in
temperature. I think its accurate to say that its practically impossible that this event would
have happened without AGW, but the attribution statement is incorrect.

Response:

As suggested by the reviewer, we have removed the attribution statement and changed the sentence to:

"Without a global warming trend, such an event would have been practically impossible."

Comment 2.4

33: was so far an extreme event
Not sure what you mean with 'so far'? Sentence is confusing

Response:

We removed 'so far' from the sentence.

Comment 2.5

34: 'return to'
I would suggest 'revert to'

Response:

Changed as suggested.

Comment 2.6

38 This jump in SSTs ...
Need to delete 'jump in', or you are saying the same thing twice. i.e you want to say that
both the sst and the jump in sst over the previous record is record breaking

Response:

Following the reviewer's comment, the sentence was changed to:

"This record-breaking SST event is not only a global record-breaking event but also unprecedented

in the magnitude by which it surpassed previous records."

Comment 2.7

41: from 2015/2016
... that occurred in 2015/2016

Response:

Changed as suggested.

Comment 2.8

43: months of a year
months of the year

Response:

Changed as suggested.

Comment 2.9

44: record-breaking jump in annual globally averaged SSTs
Is this definition specifically for the April to March period? If not how do you define this increment? Do you for example examine every 12 month (Jan-Dec, Feb-Jan, Mar-feb ...) period and compare it with all other equivalent 12 month periods?

Response:

The definition is specifically for the April to March period. Following the reviewer's comment, we have changed the sentence to:

"SSTs ("record-shattering jump" is here defined as a record-breaking jump in annual (April to March) and globally averaged SSTs that exceeds previous records by at least 0.25°C, as observed in 2023/24)"

Comment 2.10

46: , the record-shattering SSTs
You are talking about record shattering jump in SST, not record shattering SST. Its important to maintain the correct terminology

Response:

As suggested by the reviewer, we now use the correct terminology (throughout the manuscript):

"Moreover, the record-shattering jump in SSTs is believed..."

Comment 2.11

52: have stopped to be
are no longer

Response:

Changed as suggested.

Comment 2.12

Figure 1
Given you are highlighting events that beat previous records, it might be helpful to also highlight the previous record year.
If you colour the lines by year, then you could also demonstrate the gradual warming over time

Response:

We have implemented the propositions by the reviewer (Figure R1). While this new figure includes more information, we believe that the existing Figure 1 in the manuscript is more intuitive to understand and thus better transmits the essential information, i.e., the jump of the SSTs out of the group of previous SSTs. In addition, we had created Figure 1 in a way that it resembles the well-known graph from the Climate Reanalyzer that is often used in traditional and social media to present the record-shattering jump in SSTs (https://climatereanalyzer.org/clim/sst_daily/?dm_id=world2). However, we can replace Figure 1 by Figure R1 at any time if the reviewer and/or editor believes that Figure R1 fits better into the manuscript.

Comment 2.13

69: Strong increases
Large increases

Response:

Changed as suggested.

Figure 1: **Figure R1: Monthly global sea surface temperature for observations** Monthly sea surface temperature (SST) anomalies for the largest record-shattering annual (April to March) global (60°S-60°N) SST events before 2024 for observations from NOAA OI SST V2.11. The years of the record-shattering jump is shown as a blue line and the following year is shown as an orange line, and the 30 preceding years as colored lines with yellow being the year that is 30 years ago and the black line being the year before the record-shattering jump.

Comment 2.14

77 can occur

'can occur' is rather vague. If I take a random timeseries, record breaking jumps will always occur if you wait long enough (even without a warming trend)

It might be useful to list the prior record breaking jumps in order of magnitude (i.e. information from extended fig 3). In that way we can clearly see how exceptional 23/24 was

Response:

Following the reviewer's comment, we have removed the word "can".

In addition, we have adjusted the sentence and added an additional sentence to add the prior record-breaking jumps as suggested by the reviewer:

"While record-breaking jumps in global SSTs occur when a long-term warming trend is superimposed by an exceptionally warm year due to climate variability²⁶, the record-shattering jump in globally-averaged SSTs from April 2023 to March 2024 has broken previous records by a substantially larger margin than previous record-breaking jumps in SSTs. The three largest margins by which globally-averaged SSTs previously broke records were 0.16°C in 2015/16, 0.14°C in 1997/98, and 0.09°C in 2009/10."

The same information is also provided in Figure 3 in the main manuscript.

Comment 2.15

81: and whether this jump could have been expected based on fully coupled climate model simulations

This doesnt make sense as written, I think you mean ... and whether jumps of this size are simulated in climate models.

Response:

Changed as suggested.

Comment 2.16

82: If state-of-the-art fully coupled climate models were not able to reproduce such

How about: The failure of state of the art climate models to reproduce ...

Response:

Changed as suggested.

Comment 2.17

85: fully coupled
Delete? – no need to keep repeating this

Response:

Changed as suggested.

Comment 2.18

85: were > are

Response:

Changed as suggested.

Comment 2.19

86: simulate SUCH record-shattering

Response:

Changed as suggested.

Comment 2.20

86: as the one observed over the last year
Delete?

Response:

Changed as suggested.

Comment 2.21

87: and to project how SSTs continue to evolve after such events.
How about something like: ... to see if temperatures decrease again or remain high suggestive of tipping points in the climate system

Response:

Changed as suggested without adding the suggestion of a potential tipping point.

Comment 2.22

93: output from 270 climate model simulations
output from 270 simulations spanning X different climate model

Response:

Changed as suggested.

Comment 2.23

103: using an autoregressive model of order one (AR1)
I appreciate the effort the authors have gone to to include a more realistic model for SST variability. However, its still possible that the results here are sensitive to the model used. e.g. what happens if you used AR(2), ARIMA, ARFIMA ... models, each of which are based on certain assumptions about the original data? Some sensitivity testing, or at least some discussion around this topic is warranted.

Response:

Following the suggestion by the reviewer, we have tested the sensitivity of our observation-based estimate for the return period of the 2023/24 event to the statistical model used. When assuming an AR2 process instead of a AR1 process, the mean return period of 512 years (95% confidence interval 205-1185 years) changes to 484 years (95% CI of 196-1142 years). Additionally, we have tested a moving average of order 1 (MA1) model, resulting in a mean return period of 498 years (201-1179 years). Although these models differ in their autocorrelation functions, the differences in the resulting return periods are small compared to the large uncertainty from the uncertainties in the estimated autocorrelation, variance and trend of SST (as represented by the confidence intervals). We have decided to stick with the AR1 model, but now also mention the AR2 and MA1 sensitivity tests in the methods section.

We have also attempted to implemented even more complex autoregressive models. However, identifying the parameters of the more complex autoregressive model solely based on a trend, variability, and autocorrelation (lag-1, lag-2,..) becomes increasingly complicated. For some more complex models, the model parameters are not even definable for certain combinations of sampled parameters (e.g., ARMA(1,1)). Thus, we have decided to test the sensitivity of the results based on the AR1, AR2, and MA1 autoregressive models for which all parameters are clearly defined.

Comment 2.24

107: under the current long-term warming trend

This relatively short period that is used to calculate the warming trend, will be sensitive to the internal variability (or even external factors like volcanoes). Doesn't this present an important uncertainty?

Using your large ensembles you can actually estimate the forced warming (using the ensemble average) and examine the degree of uncertainty associated with individual ensemble member. However you can't do this with the observations.

I'm not sure how best to handle this, but it should be given some careful consideration

Response:

The linear trend over a relatively short time period is indeed very sensitive to internal variability. We had represented the resulting uncertainty based on the expected uncertainty of such trends in timeseries where the residuals from the trend are given by Gaussian white noise (Wilks, 2019; and please see Methods and Extended Data Figure 7). Following the reviewer's comment, we have now given additional consideration to the uncertainties related to the trend, revised the best estimate of the trend and its uncertainty accordingly, re-estimated the return periods, and added the following text to the Methods in the revised manuscript:

"The underlying warming trend in SSTs in 2023/24 was estimated using an Enting spline³⁹ that is fitted to the observation-based globally and annually averaged SSTs from 1982/83 to 2023/24. From this spline fit, the trend in 2023/24 was calculated as the slope of that spline fit in 2023/24. The Enting spline filters short-term variability from timeseries with noise. The amount of noise that is removed depends on the cut-off period, with low cut-off periods removing only the short-time variability and large cut-off periods removing both the short-term and longer-term variability. To determine the cut-off period that allows a robust estimate of the trend in 2023/24, the trend in the 11 climate simulations that include a record-shattering jump in SSTs was calculated for cut-off periods ranging from 25 to 55 years. Cut-off periods below 25 years do not filter the decadal variability, and cut-off periods of 55 years are too rigid to capture non-linear components⁴⁰ of long-term warming trends. In each simulation with a record-shattering jump, the Enting spline was fitted to the year of the jump and the 41 years before. The length of 42 years was chosen as NOAA OI SST V2.1¹ also covers 42 years. The so-estimated trend was then compared to the 'true' underlying trend. This 'true' trend was estimated with a 31-year running mean that can be calculated for the jump years in models because the SSTs after the jump are also simulated by the models. An Enting spline with a cut-off period of 40 years fits that underlying 'true' trend best and is only $4\pm 12\%$ larger, i.e., $0.01\pm 0.03^\circ\text{C}$ per decade). With this approach applied to NOAA OI SST V2.1¹, we estimate the trend in 2023/24 to be 0.269°C per decade. The uncertainty of the estimated trend was quantified using the synthetic timeseries of 100 million years with a trend of 0.269°C per decade and the best estimate of the variance and autocorrelation (see paragraph below). At each of the around 225,000 record-shattering jumps in this timeseries, the trend was estimated with an Enting spline with a cut-off period of 40 years. The standard deviation of these estimated trends is 0.037°C per decade. Thus, the estimated trend from the Enting spline has a likelihood of 95% to be between 0.195 and 0.343°C per decade.

To further quantify the sensitivity of the resulting trend estimate to the estimation method, we

fitted a linear trend from 2004/2005 to 2023/24 to approximate the trend in the year of the jump in NOAA OI SST V2.1¹. The slope of this linear fit is 0.254°C per decade, slightly smaller than the estimate from the Enting spline (0.269±0.037°C per decade) as it does not capture the slight non-linear warming component⁴⁰ but well within the uncertainty range. These sensitivity tests highlight that the largest uncertainty of the trend estimates results from the climate variability that is superimposed on the underlying trend and not from the method that is used to determine the trend."

We have further repeated the analysis by calculating the 'true' underlying trend using an Enting spline with a wide-range of cut-off periods over the entire timeseries before and after the jump. Also in these tests, the best fit between the 'true' underlying trend is found when fitting an Enting spline with a cut-off period of 40 years to the 42 years before the jump in global SSTs. Although this analysis further corroborates our results, we have not added this additional analysis to the manuscript as we fear that it would confuse the reader and as our uncertainty range for the estimated trend already covers the difference in trend estimates when using different cut-off periods or even a linear fit.

Wilks, D. S. Statistical Methods in the Atmospheric Sciences. (Elsevier, 2019). 505 doi:10.1016/C2017-0-03921-6.

Comment 2.25

110: Without human-made global warming, however, a record-shattering jump in globally averaged SSTs as observed last year would not have been possible

I think the statement is not quite correct. The trend that you are imposing isnt all related to anthropogenic forcing. I think it would be more correct to say without a strong long term warming, the chance of a record-shattering jump becomes negligible. Irrespective of the variability or autocorrelation characteristics, without a trend we found no record-shattering jumps in our synthetic timeseries.

Response:

We have adjusted the sentences following the suggestions by the reviewer:

"Without underlying warming, the chance of a record-shattering jump becomes negligible. We found indeed no record-shattering jumps in our synthetic timeseries without a long-term warming trend, irrespective of the variability or autocorrelation characteristics (Extended Data Figure 2)."

Comment 2.26

119: years between 2000 and 2040

delete - you have just said this

Response:

Changed as suggested.

Comment 2.27

123: 95% confidence interval; see Methods

How do you calculate the confidence interval in the models. You only have one set of 11070 years from which you count the number of record jumps

Response:

The estimation of the confidence interval was already described in detail in the Methods. We had addressed a similar comment by the same reviewer (comment 2.45 in the last review) by adding more details about the estimation of the confidence intervals in the Methods. Hence, we are not sure what additional information is needed and would be grateful if the reviewer could point to the point of misunderstanding so that we can clarify the explanation. In addition, we added additional information to the main manuscript and hope that this information will improve the understandability here:

"...event a 1-in-1,000-year event in climate models (1-in-550-year to 1-in-2,000-year event; 9% confidence interval using the Pearson-Clopper confidence interval for binomial experiments; see Methods) (Fig. 3b)."

Comment 2.28

123: Even jumps in globally averaged SSTs that break the 124 previous record by a larger magnitude than 0.25C are not impossible in climate model simulations but increasingly unlikely (Fig. 3b).

How about: There are a small number of jumps that exceed 0.25, but the probability of a certain jump magnitude reduces with magnitude. For example jumps that exceed 0.20C are 6-7 time more likely than jumps exceeding 0.25oC.

Response:

Following the reviewer's comments, we changed the sentences to:

"Overall, the probability of a simulated record-breaking jump in SSTs with a certain magnitude reduces with that magnitude. For example, jumps that exceed 0.2°C are 6-7 times more likely than jumps exceeding 0.25°C, while jumps that exceed the previous record by a larger magnitude than 0.25°C become less likely but are not impossible in climate model simulations (Fig. 3b)."

Comment 2.29

128-131: How about: Under high emissions scenarios the probability of record shattering events increases, in line with higher rates of warming. However under strong mitigation scenarios (S3A), the warming trend reduces in the future and there are no simulated record shattering events simulated across the model ensemble (S3B).

Response:

Following the reviewer's suggestion, we have changed the sentences to:

"Under high emissions scenarios, the probability of record shattering events increases (Extended Data Figure 3a), in line with higher rates of warming. However, under strong mitigation scenarios, the warming trend reduces in the future and there are no simulated record shattering events simulated across the model ensemble (Extended Data Figure 3b)."

Comment 2.30

139: not only

This is not a proper sentence. I think you just need to delete 'not only' .

Response:

We believe that there has been a misunderstanding by the reviewer. 'Hot models' do not simulate a higher rate of record-shattering jumps in SSTs. We have changed the sentence to clarify the apparent misunderstanding:

"Record-shattering global jumps in SSTs ($>0.25^{\circ}\text{C}$) are simulated by different models with a wide range of trends, autocorrelations, and variabilities. It is thus not only the so called 'hot models'³² with high transient climate responses, like IPSL-CM6A-LR, CESM2, and CanESM5 that simulate such events."

Furthermore, we ended the paragraph with the following sentence for further clarification:

"Overall, no model stands out with an unusually small or high number of extreme events per number of simulated years (see Methods)."

Comment 2.31

139: 'hot models'

Have you done this analysis taking into account the fact that different models have different numbers of years? i.e. your statement should be something like: climate models that have the highest rate of record shattering events at the hot models

Response:

Following a comment by the same reviewer in the previous round, we had already added such an analysis to our responses. We have now added it also to the Methods:

"We have not detected an unusual proportion of record-shattering SST jumps in any of the model ensembles. Therefore, most record-shattering jumps in SSTs are found in model simulations that tend to have the largest number of ensemble members (2 such events in 30 GFDL-ESM2M-LE members, 2 in 50 CESM2-LE members, 2 in 50 CanESM5 members, 4 in 43 MIROC6 members, and 1 in 6 IPSL-CM6A-LR members). Given that the observation-based estimate of the return of

record-shattering jumps in SSTs has a confidence interval between a 1-in-a-205-year event and a 1-in-a-1,185-year event, it appears plausible that no such events are sampled in model ensembles with 10 or less ensemble members, which cover a maximum of 410 years. The smallest ensemble in which an event was simulated was IPSL-CM6A-LR, which runs over 246 years. Finding such an event in one small ensemble is also plausible given that there are many such small ensembles without a record-shattering SST jump. Lastly, the most events were found in the MIROC6 ensemble with 4 events in 1,763 years (43 ensemble members). This corresponds to a return period of around 440 years, within the uncertainty range of the observation-based estimate."

Comment 2.32

140: Models with
However model with ...

Response:

Following the reviewer's comments, we have added a word to link the two sentences. However, instead of adding "however" as suggested by the reviewer, we have added "instead" as we believe that it reflects the meaning of the sentence better and helps to avoid the apparent confusion around the 'hot model' paragraph.

Comment 2.33

143: In general, such extreme events are simulated by models that provide large ensembles
I dont understand this. Of course models with lots more years are much more likely to have more events. This doesnt tell us anything useful. You need to make like for like comparisons. If you wanted to do this you could subsample so that all models have the same number of years, or you could create long synthetic simulations based on each model.

Response:

We have removed the sentence from the manuscript. Instead we write:

"Overall, no model stands out with an unusually small or high number of extreme events per number of simulated years (see Methods)."

We further would like to refer the reviewer to the more detailed analysis that was already in the Methods and is cited in response to comment 2.31.

Comment 2.34

168: the record-shattering jump in SSTs
Its confusing saying this here as there was no jump in 2023/24 on this region. Just say SSTA was once larger than in 2023/34. The same applies to the eastern tropical pacific below

Response:

As suggested by the reviewer, we now say that SSTs were once larger than in 2023/24 and do not talk about a jump in these regions.

Comment 2.35

169: In climate models, events as large as that in 2023/24 all coincide with an El-Niño event, i.e., a positive El Niño-Southern Oscillation phase of at least 1.5°C in the El-Niño 3.4 index (Fig. 4b-d).

Fig 4 only shows 3 model events. Are you saying this is consistent across all such events?

Response:

It is indeed consistent across all events. We now clarify this as follows:

"In climate models, all 11 jumps in globally-averaged SSTs that are as large as that in 2023/24 coincide with an El-Niño event, i.e., a positive El Niño-Southern Oscillation phase of at least 1.5°C in the El-Niño 3.4 index (Fig. 4b-d)."

Comment 2.36

189: the simulated 11
the 11simulated. . .

Response:

Changed as suggested.

Comment 2.37

193: is 0.09°C smaller than the extraordinarily large observed 193 margin of 0.42°C unnecessary to give both the 0.09 and the 0.42. One of these numbers will do

Response:

As suggested, we have removed the second number.

Comment 2.38

194 large regional
delete regional - unneeded as you say North Atlantic

Response:

Changed as suggested.

Comment 2.39

194: in the 270 climate model simulations

Its more useful to talk about the number of years (11,000) than the number of simulations

Response:

Changed as suggested.

Comment 2.40

197: An even larger model ensemble would be necessary to estimate the likelihood of the combination of a global record-shattering jump in SSTs and a record-shattering jump in SSTs in the North Atlantic in 2023 and 2024.

Not really. You could use exactly the same approach as you have done for the observations. Also Im not really sure why we specifically care about this number, unless you are suggesting some specific mechanism that links global to NA SST

Response:

The size of the jump in the North Atlantic is important information because the North Atlantic SST broke records by an extremely large margin. These exceptional North Atlantic SSTs have caught attention by the scientific community and the media. Hence, we feel that it is important to address this particular region in a manuscript about record-shattering SSTs globally. Following the suggestion by the reviewer, the sentence was corrected to:

"An even larger model ensemble would be necessary to see if climate models can simulate the combination of a global record-shattering jump in SSTs and a record-shattering jump in SSTs in the North Atlantic in 2023 and 2024."

We have also considered the suggestion to estimate the theoretical return period of a simultaneous record-shattering jump in global and North Atlantic SSTs. To estimate the return period of a global record-shattering event with co-occurring record-shattering in the North Atlantic based on a statistical model, as proposed by the reviewer, would require modelling the statistical relationship between global and NA SST. As this relationship is very uncertain and as the individual return periods are also highly uncertain, we would only be able provide very vague results. We have thus decided to not do this analysis

Comment 2.41

202-206: Repetitive

I find this NA analysis rather long and not very well motivated. Unless you could say for example that without the large jump in the NA then 2023/24 wouldn't have been a particularly noteworthy global event. For example you might show the temperature anomaly relative to 2016/17 to show which areas contributed most to the record shattering increase.

Response:

We have removed the repetitive part as suggested. In the response to the previous comment, we have explained why we kept the paragraph about the North Atlantic although high North Atlantic SSTs are not necessary for a large jump in SSTs globally. We also made the suggested map (Figure R2c). It shows high jumps in the North Atlantic and the North Pacific as well as around the Southern Ocean. However, this map might be misleading as the North Pacific has previously been warmer than in 2023/24 but not in the year of the previous record 2015/16. In addition, the large differences between 2023/24 and 2015/16 (Figure R2c) do not necessarily result from large SSTs in 2023/24 (Figure R2a) but might result from low SSTs in 2015/16 (Figure R2b), like in the subpolar North Atlantic or the western North Pacific. As it is very difficult to extract additional information about the underlying mechanisms of high SSTs in 2023/24, and as we have reduced the paragraph about the North Atlantic to a minimum in response to the reviewers suggestion, we did not add the suggested figure to the manuscript.

Comment 2.42

208: SST anomalies develop

SST typically anomalies develop

Response:

Following the reviewer's suggestion, we have changed the sentence to:

"We now use these model simulations to understand how SST anomalies typically develop over the years that follow a record-shattering jump in SSTs."

Comment 2.43

209: delete 'such'

Response:

Changed as suggested.

Figure 2: **Figure R2: Sea surface temperature anomalies for the record-shattering jump in 2023/24, the previous record in 2015/16 and the difference between both.** Monthly sea surface temperature (SST) anomalies **a)** for the largest record-shattering annual (April to March) global (60°S-60°N) SST events before 2024 and **b)** for the previous record in 2015/16, as well as **c)** the difference between both. All anomalies are calculated from NOAA OI SST V2.1. All anomalies are calculated with respect to 1993/94-2022/23.

Comment 2.44

210: stop to be
... stop being

Response:

Changed as suggested.

Comment 2.45

212: have also stopped to be record breaking in
dropped below record levels

Response:

Changed as suggested.

Comment 2.46

214: to their level before the 214 jump
return to their level before the 214 jump

Response:

The sentence in line 214 already was "return to their level before the jump", the sentence that was suggested by the reviewer. We thus kept it as is.

Comment 2.47

216: In only
However ...

Response:

Changed as suggested.

Comment 2.47

217: do not return to pre-jump levels

I can see this is true in panels c) and d) as the subsequent 10 years are all warmer than pre jump years. Is the other one f)?

You should explicitly explain what you mean by this, i.e. monthly global SST anomalies remain above all the same months in pre-jump years for the subsequent 10 years

Also its helpful to refer to specific panels in your figure

Response:

We have now changed the sentence according to the suggestions by the reviewer:

"However, in 3 of the 11 events, the SST anomalies do not return to pre-jump levels over the next 10 years (Extended Data Figure 4 b-d) and beyond (Extended Data Figure 5), so that the global SSTs have indeed permanently risen to a higher level in these simulations."

Comment 2.48

218: simulations. However, even in these three cases, the warming trend after the jump in SSTs has not accelerated as these models' SSTs warm less fast in the years following the jump and return to the respective ensemble mean within 8 years after the event (Extended Data Figure 5).

I find the statement confusing. I think you are saying is that after the event temperatures revert back to the long term trans over the course of a few year i.e. there's no shift to a new warming trajectory

Response:

Following the reviewer's suggestion, the sentence was changed to:

"However, even in these three cases, SSTs revert back to the expected long -term warming trend over the course of at most 8 years and do not shift to a new higher or steeper warming trajectory (Extended Data Figure 5)."

Comment 2.49

224: significantly

Only use the word significantly if you mean that the difference is statistically significant.

Response:

Following the reviewer's suggestion, we have replaced 'significantly' by 'substantially'.

Comment 2.50

224: rate32, facilitating continuing high levels of SST anomalies after the record-shattering jump in SST

Its ocean temperatures that will primarily set air temperatures over the ocean not the other way around

Response:

We have now clarified this in the revised manuscript as follows and added an appropriate reference (Deser et al., 2009):

"The three cases occur in CanESM5 and IPSL-CM6A-LR, two climate models with extremely high transient climate responses³⁴ outside the recently assessed range³⁵ and with atmospheric warming rates from 1981 to 2014 that substantially exceed the observed rate³⁴. As atmospheric warming and sea surface warming are generally strongly linked³⁶, the also too high SST warming rates facilitate continuing high levels of SST anomalies after the record-shattering jump in SSTs."

³⁴Deser, C., Alexander, M. A., Xie, S.-P. & Phillips, A. S. Sea surface temperature variability: patterns and mechanisms. *Annu. Rev. Mar. Sci.* 2, 115–143 (2009).

Comment 2.51

225: As SST anomalies only do not return to pre-jump levels in these 'hot models'³⁰, it is likely

Given that a return to pre-jump temperatures only fails to occur in these hot models ...

Response:

Changed as suggested.

Comment 2.52

227: to where they were
temperatures mote typical of those...

Response:

Changed as suggested.

Comment 2.52

227: in globally averaged SSTs
Delete

Response:

Changed as suggested.

Comment 2.53

228-234: Hard to follow. I suggest you stick to what your analysis is actually showing you. I think the main thing you need to say is that... Only under much larger warming rates than observed [and I suspect lower internal variability] might we expect temperatures to consistently remain above pre-jump years. Moreover, in the coming months we can expect to see temperatures revert back towards the long-term trend

Response:

We have clarified the sentence following the reviewer's suggestions:

"If, however, observed SSTs do not return to pre-jump levels by September 2025, we expect SSTs to revert back towards the long-term trend over the following 8 years (Extended Data Figure 5). If this is also not the case, the ongoing extreme event would not be consistent with climate model simulations."

We have kept the second part that describes when the event would not be consistent with climate model simulations anymore because it was of great importance to reviewer 1.

Comment 2.54

237: a 1-in-a-760-year
I dont think providing a single number without uncertainty bounds is useful. Maybe just say 'a very rare'

Response:

We prefer to remain quantitative here and thus did not change "a 1-in-a-760-year" to "a very rare", but decided to opt for the second option proposed by the reviewer and added uncertainty bounds. We did, however, update the return period based on our revised estimate (please see response to comment 2.24).

Comment 2.55

239-242: This sentence is confusing. Break into 2 or 3 sentences. Also its only some models that simulate these events, but they are very rare,. And they are more common in models that have faster rates of anthropogenic warming

Response:

As suggested by the reviewer, we have now broken the sentence in 2 sentences:

"We have further shown that climate models indeed simulate such global (60°S-60°N) annual record-shattering jumps in SSTs that exceed the previous records by 0.25°C, like the global jump in SSTs that was observed last year. Moreover, the estimated return period of these events in climate models (1-in-a-1,000-year events in models) is within the confidence interval of the observation-based estimate of the return period."

Furthermore, we believe that there was a misunderstanding with respect to such events being more common in models with a faster warming rate. There is no such bias and we have addressed this misunderstanding in the responses to comments 2.30 and 2.31.

Comment 2.56

242: Furthermore,
Furthermore, in these models, ...

Response:

Changed as suggested.

Comment 2.57

243: stop to be record breaking
drop below record levels

Response:

Changed as suggested.

Comment 2.58

245: have stopped to be
Stopped being

Response:

Changed as suggested.

Comment 2.59

248: warm slower over the following decade and return to the expected warming trajectory. I find this statement ambiguous. This seems to imply the AGW slows down for some years. Its simply that temperatures revert to the long term mean warming trend over the span of a few years, as would be expected if this event was just related to internal variability (albeit a very rare jump)

Response:

Following the reviewer's suggestion, the sentence was changed to:

"In the few simulations that do not simulate a return to pre-jump levels, SSTs revert to the expected warming trajectory over the following years."

Comment 2.60

248-253: Im not sure it gives me great confidence - you only find these events in a subset of models. What provides confidence is if the model can simulate accurate warming rates and has variability that matches observations across all timescales. You wouldn't validate a model based on a single extremely rare event

Response:

We believe that there is a misunderstanding. We have shown that different models over a wide range of different trends, variabilities, and autocorrelations simulate record-shattering jumps in SSTs with no model having an unexpected low or high rate of such jumps (Extended Data Figure 3 and responses to comments 2.30 and 2.31.). It is not not only simulated in a subset of models with large warming rates. We therefore keep this sentence unchanged.

Comment 2.61

349: within +/-10%.
In S7C you have one autocorrelation at 0.2 and the other at 0.4. This is not a <10% difference. The same is true for the extreme values of the SST variance 0.0067 and 0.0083 are more than 10% different, so Im not clear what you mean here

Response:

The range of $\pm 10\%$ refers to the uncertainty related to the choice of the fit that is used to detrend the SST timeseries and not to the overall uncertainty of the variability or autoregression, as described in the same paragraph:

"The resulting uncertainties are relatively large (Extended Data Figure 7b, c) as the internal climate variability still renders the estimation of the actual variability and autocorrelation uncertain

in a timeseries that covers 84 years from 1940 to 2023. These large uncertainties due to the internal variability cover the individual estimates from the three reanalysis products (Extended Data Figure 7). Furthermore, the sensitivity tests towards the detrending method were evaluated. When using a 2nd order polynomial or an Enting spline with a cut-off period of 40 years, the results change by $\pm 10\%$ and are within the large uncertainties in the estimates resulting from the internal climate variability."

We would also like to refer to our response to the reviewer to comment 2.42 in the last review round that addresses this subject in detail and led to the sensitivity test with different detrending methods.

Comment 2.62

351: Extended Data Figure7

I dont understand what is being presented in this figure. You have 3 data products and 3 detrending methods, where does the probability distribution come from?

In S7A what does the trend value represent - a 3rd order polynomial trend isnt represented by a single number?

Why are you using a single product in panel A and 3 products for the other parameters?

In C why can we only see lines for 2 data products.

Using colours rather than different linestyle would be much clearer

Response:

In response to the reviewer, we have largely re-organized and re-written the Methods section. Here, we additionally provide responses to each of the questions:

For the trend, we have largely revised the way we estimate its size and the underlying uncertainty. These changes are explained in detail in our response to comment 2.24. In addition, we now also explain why we have chosen to rely on only one observation-based product in our analysis for the trend estimate:

"Observed SST anomalies were calculated using the global and highly-resolved NOAA OI SST V2.1¹ product, which is based on observations from satellites, ships, buoys, and ARGO floats. Among the different available SST products³⁷, we have chosen NOAA OI SST V2.1¹ as it has shown the best performance³⁷. In addition, NOAA OI SST V2.1¹ is also the only dataset that includes the observations from the ARGO program that was started in 1999 and operates between 3,000 and 4,000 ARGO floats since 2007."

To test the sensitivity with to the choice of the dataset, we now also repeated the analysis with a second observation-based SST database, which also performs well but not as well as NOAA OI SST V2.1, and added the following text to the Methods in the revised manuscript:

"To test the sensitivity to the choice of the observation-based SST estimate, the underlying warming trend, the magnitude by which the record-shattering jump in SSTs in 2023/24 broke the previous SST record, and the return period of such record-shattering jumps in SST was also quantified using the SST Analysis production version 3.0 from the European Space Agency Sea Surface Temperature Climate Change Initiative based on the OSTIA reanalysis system ICDR3.0³⁸. In that dataset, the

SST jump in 2023/24 is 0.23°C, the underlying trend in 2023/24 is 0.209°C. per decade, and the resulting return period estimate is 543 years with a 95% confidence interval of 204 to 1,371 years. Thus, the return period is almost the same as the return period estimated based on NOAA OI SST V2.1¹. Furthermore, the number of record-shattering events in climate models for a jump of 0.23°C is 19, resulting in a return period of 583 years. Thus, the estimates of the return period of record-shattering events based on observations and models are closer when using this dataset instead of NOAA OI SST V2.1¹.."

For the autocorrelation and variance, longer timeseries than the timeseries from the satellite observations are preferred to reduce uncertainties. To further reduce uncertainties in these estimates, we here use the average of three reanalysis products. The mean estimates over the three products are then used as the best estimate for the autocorrelation and variability. We have further extended our explanation about the estimation of the uncertainty around these best estimates:

"As opposed to the estimate of the trend, we could not rely on observation-based SST estimates over the satellite period, like NOAA OI SST V2.1¹, as the relatively short length of these estimates (1982 until today) makes uncertainties of the autocorrelation and variance large. Instead, the observation-based variance and lag-one autocorrelation are estimated from the three SST reanalysis products described above over the period 1940-2023 after detrending the data using a cubic spline, resulting in three estimates each. As it is not evident which reanalysis product performs best, the mean of all three estimates is used as the most likely estimate of the autocorrelation and variability. An implicit assumption for the estimation of the autocorrelation and variance is that the temperature variability follows a Gaussian distribution (Extended Data Figure 6). For the ERA5, HadISST, and ERSST reanalysis products, three tests (Kolmogorov Smirnov, Shapiro-Wilk, and Anderson-Darling) were employed to test whether the SST anomalies follow a Gaussian distribution. Across these three tests and three timeseries the p-values vary from 0.26 to 0.90 indicating no significant deviation from a Gaussian distribution. We thus conclude that the SST data is well modelled by a Gaussian distribution. To estimate the uncertainty of the autocorrelation and variance, sampling distributions for the best estimates of the variance and autocorrelation are constructed. For normally distributed time series of a given length, theoretical sampling distributions of variance and autocorrelation estimates are known^{41,42}. These sampling distributions characterize the dispersion of estimates around a true value. The variance of the sampling distribution thus informs about the uncertainty in the estimate from internal climate variability for an estimate from a single product. As the best estimates of the variance and autocorrelation are both an average of three products, the uncertainty is smaller than the uncertainty from a single reanalysis product. To reflect this reduced uncertainty, we randomly sample 10,000 times three values from the respective distribution and average over these three values. The resulting values are then the sampling distribution of the average estimate of the three reanalysis products. The resulting uncertainties are relatively large (Extended Data Figure 7b, c) as the internal climate variability still renders the estimation of the actual variability and autocorrelation uncertain in a timeseries that covers 84 years from 1940 to 2023. These large uncertainties due to the internal variability cover the individual estimates from the three reanalysis products (Extended Data Figure 7). Furthermore, the sensitivity tests towards the detrending method were evaluated. When using a 2nd order polynomial or an Enting spline with a cut-off period of 40 years, the results change by $\pm 10\%$ and are within the large uncertainties in the estimates resulting from the internal climate variability."

In addition, we added color-coded lines in Extended Data Figure 7. One line was hidden behind the mean estimate. After correcting for a minor mistake in the calculations, the estimate has

changed and is not anymore hidden behind the mean.

Comment 2.63

374: were calculated from April to March

Im not really sure about the use of April to March. In the climate models you might find larger increments in other months. Also is the April - March jump actually the largest in the observations e.g. compared to May-April, Jun-May etc?

Response:

We had chosen the time period from April to March as the SSTs started to separate visually from the previous record in April 2023 (Figure 1a). Differences to previous records are up to 0.009°C larger when averaging from March to February, May to April, June to May, and July to June.

In climate models, the number of events when SST jumps break the previous record by at least 0.25°C also varies by month:

Jan-Dec: 4 times
Feb-Jan: 3 times
Mar-Feb: 6 times
Apr-Mar: 11 times
May-Apr: 11 times
Jun-May: 11 times
Jul-Jun: 12 times
Aug-Jul: 11 times
Sep-Aug: 11 times
Oct-Sep: 10 times
Nov-Oct: 5 times
Dec-Nov: 3 times

Overall, such jumps mostly start between April and October. Within this timespan, the choice of months over which one averages does not appear to substantially affect the results.

As this manuscript specifically treats the observed jump in SSTs, which started in April, we decided to keep the annual average from April to March. However, we added the following sentence to the methods to transparently show the effect of changing the months over which the annual means are calculated:

"Here we chose to calculate annual averages starting in April as this is the month when SSTs started to break records by a larger margin (Figure 1a). However, differences to previous records in NOAA OI SST V2.1¹ are up to 0.009°C larger when averaging from May to April, June to May, and July to June, and become smaller afterwards. The number of times that such events occur in climate models varies between 11 and 12 when the first month over which one calculates the annual average is between April and July. The results are hence similar for averaging periods starting later in the year."

Comment 2.64

377: Most record-shattering jumps in SSTs are simulated by models that tend to have the largest number of ensemble members

Isn't this a trivial statement? More years so more events. Some account should be made for the overall length of data from different models

Response:

Following the reviewer's comment, we now write:

"We have not detected an unusual proportion of record-shattering SST jumps in any of the model ensembles. Therefore, most record-shattering jumps in SSTs are found in model simulations that tend to have the largest number of ensemble members..."

The overall length of data is presented in Extended Data Table 1.

Comment 2.65

387: models is 20% smaller

I presume you mean the multi-model mean trend? Always more useful to provide a range (e.g. interquartile range)

Response:

Following the reviewer's comment, we have added "multi-model mean" to the manuscript and now provide the range by adding the interquartile range.

Comment 2.66

387: 2004 to 2023

I don't really understand the justification for using a (linear?) trend from 2004-2023. This will be highly contaminated by internal climate variability. For example if you take one of your large ensembles and look at the differences in trends between members for 2004-2023 the differences will likely be quite large. Alternatively, if you just calculate the confidence interval on the trend it would be quite large. I think for a mean of 0.25°C/decade the uncertainty would probably be from 0.2 to 0.3 °C/decade

Response:

In response to the reviewer's comment, we have largely revised the way we determine the underlying trend and its uncertainty. The new estimate is 0.269 ± 0.037 °C per decade. Thus, there is a 95% likelihood that the trend is between 0.195 and 0.343 °C per decade. This large range highlights the large uncertainty. This new estimate is not significantly different from our previous estimate based on the linear trend, which was 0.25 ± 0.03 °C per decade and covered a 2-sigma range from 0.19 to

0.31°C per decade. For more details, please see our response to comment 2.24.

Comment 2.67

395: addition, the autocorrelation in CMIP6 models is 55% you are presumably referring to the multi-model mean. This makes it sound like all models are 55% higher. These descriptions need to be more precise

Response:

Following the comment by the reviewer, we now precise that these are multi-model means.

Comment 2.68

401-408: This information should go after describing the idealised timeseries

Response:

During the re-ordering of the Methods section, this information has been moved so that it appears after describing the idealized timeseries as suggested.

Comment 2.69

428:, the Kolmogorov Smirnov Shapiro-Wilk or Anderson-Darling tests are much more typically used to test for data normality, why have you used a KS test?

Response:

As suggested, we have repeated the tests for normality with the Shapiro-Wilk and Anderson-Darling tests. We find similarly high p-values and no significant deviations from a Gaussian distribution for all three reanalysis products. We have added the results to the manuscript.

Comment 2.70

436: observational trend is estimated over the period 2004-2023 using the NOAA OI SST V2.11
Why three estimates of variance and autocorrelation and only one estimate of trend?

Response:

For the trend, we have only used the NOAA OI SST V2.1 observation-based estimate as this estimate performed best when evaluated against independent data and also because it is the only product that includes data from ARGO floats (Huang et al., 2023; <https://journals.ametsoc.org/view/journals/atot/40/4/JTECH->

D-22-0081.1.xml). We have, however, now tested the sensitivity to the choice of the observation-based product by estimating the trend, jump magnitude, and return-period of record-shattering jumps with another observation-based product that also performs well but not as well as NOAA OI SST V2.1.

For the autocorrelation and variability, we could not only rely on observation-based estimates but had to use reanalysis products, which provide timeseries that extent beyond the limited satellite period. As we are not aware that one of these products performs better than another, we decided to use the mean estimate of these products.

For a more detailed answer and associated changes to the manuscript, we refer the reviewer to our response to comment 2.62.

Comment 2.71

432-226: I find this description rather hard to follow. I think you could explain this more clearly, starting with your goal.

E.g. In order to obtain an estimate of the uncertainty in the observational return frequency estimate of record shattering events (i.e. $>0.25^{\circ}\text{C}$ jump), we randomly sample trend, variance and autocorrelation estimates based on ... 10,000 times. 10,000 AR1 models are subsequently constructed and the return period is calculated.

Response:

The Methods have been largely re-structured. During this process, we have implemented the reviewer's suggestion as follows:

"To estimate an observation-based estimate and the associated uncertainties of the return period of record shattering SST events that breaks the previous record by the margin of 2023/24, synthetic timeseries of lengths of 100 million years were constructed."

2 Response to reviewer 2

General Comments

This is a re-review of Likelihood of record-shattering jump in sea surface temperatures in 2023/24

Again I thank the authors for diligently responding to my new set of questions and comments. I have only minor suggested edits below.

One final general point is that there are now at least two studies I have seen describing some of the factors that attempt to explain the global or regional SST jumps in 2023:

<https://www.science.org/doi/10.1126/science.adq7280>

<https://www.nature.com/articles/s43247-024-01413-8>

It might be worth including a short description of the proposed mechanisms in the manuscript.

Regards

Alex Sen Gupta

Response:

We thank the reviewer for their positive evaluation and the additional helpful comments. We have integrated the two above-mentioned references, have taken each comment into account, provide responses to each point below, and adapted the manuscript accordingly.

Comment 2.1

18: Global ocean surface temperatures have been at record levels . . .

Since about July 2024 they were no longer at a record level. So 'have been' should probably now be 'were'

Response:

Changed as suggested.

Comment 2.2

22: of this still ongoing

As above this is no longer the case (sorry that's probably because of my long reviews!)

Response:

'Still ongoing' was removed as suggested.

Comment 2.3

44: coincides in time with atmospheric record warming
'Warming' implies that dT/dt was at a record. Do you mean: coincides in time with record atmospheric temperatures?

Response:

Changed as suggested.

Comment 2.4

75: superimposed by
superimposed onto

Response:

Changed as suggested.

Comment 2.5

91: Here we use observation-based monthly-mean SST estimates from various data products (NOAA OI SST V2.11, ERA5²⁹, HadISST³⁰, ERSST³¹)
Here we use observation-based monthly-mean SST estimates from various data products (NOAA OI SST V2.11 , HadISST³⁰, ERSST³¹) and reanalysis (ERA5²⁹) ...

Response:

The sentence was changed to:

"Here we use observation-based monthly-mean SST estimates from various data and reanalysis products (NOAA OI SST V2.1¹, ERA5³⁶, HadISST³⁷, ERSST³⁸)..."

The numbering of the references has changed during the revisions.

Comment 2.6

92: analyze the output from 270 simulations
analyze the output from 270 combined historical and future simulations

Response:

Changed as suggested.

Comment 2.7

133: over a finite timeseries
over a short timeseries

Response:

Changed as suggested.

Comment 2.8

134: so that the return period of such events might well be 1000 years as estimated from the models
I don't think this statement is useful. It might well also be 100 years. I would just delete

Response:

We believe that this sentence is important as it compares the two estimates of the return period, the one from idealized timeseries and the one from climate models. Therefore, we prefer to keep this sentence in the manuscript.

Comment 2.9

In Extended Data Table 1 there are some strange numbers (757,758,759) in large font near the end of the table

Response:

The numbers are an artifact of changing the Word file into a PDF file. The table has been adjusted in response to an editorial request so that these numbers are not occurring in the table anymore.

Comment 2.10

235: the also too high SST
... the overly high SST ...

Response:

Changed as suggested.

Comment 2.11

240: over the following 8 years. . .
... within a few years ...

Response:

Changed as suggested.

Comment 2.12

241: If this is also not the case
If this were not the case ...

Response:

Changed as suggested.

Comment 2.13

244: Based on observations of past SST and using an AR(1) model
How about: Based on long synthetic timeseries with temporal characteristics that match
available observations ...

Response:

Changed as suggested.

Comment 2.14

246: in the context of human-induced climate change to date
Maybe: ... based on current warming rates ...
I would also break to 2 sentences. i.e. Such a jump would not have been possible without ...

Response:

Changed as suggested.

Comment 2.15

249: by 0.25°C
... by at least 0.25°C

Response:

Changed as suggested.

Comment 2.16

364: from the three reanalysis products ERA529, HadISST30, and ERSST31
Of these only ERA5 is a reanalysis (i.e. a model with data assimilation). The others are interpolated observational products

Response:

The sentence was changed to:

"In addition, longer observation-based SST timeseries from one reanalysis product (ERA5³⁶) and two interpolated observational products (HadISST³⁷, and ERSST³⁸) from 1940 to 2023"

Comment 2.17

414: trend estimate to the estimation method
... trend to the estimation method

Response:

Changed as suggested.

Comment 2.18

428: As it is not evident which reanalysis product
As above, only one of these is a reanalysis product

Response:

The sentence was changed to:

"As it is not evident which product performs best,..."

We have also gone through the entire manuscript to make sure that the products are described in the correct way.